# SURGE: Surrogate Gradient Adaptation in Binary Neural Networks

**Haoyu Huang** [1] [*]   **Boyu Liu** [2] [*]   **Linlin Yang** [3] [‡]   **Yanjing Li** [4]   **Yuguang Yang** [4]   **Xuhui Liu** [5]   **Canyu Chen** [1]
**Zhongqian Fu** [6] [†]   **Baochang Zhang** [2] [7]

## Abstract

The training of Binary Neural Networks (BNNs) is fundamentally based on gradient approximation for non-differentiable binarization operations (*e.g.*, `sign` function). However, prevailing methods including the Straight-Through Estimator (STE) and its improved variants, rely on handcrafted designs that suffer from gradient mismatch problem and information loss induced by fixed-range gradient clipping. To address this, we propose SURrogate GradiEnt Adaptation (SURGE), a novel learnable gradient compensation framework with theoretical grounding. SURGE mitigates gradient mismatch through auxiliary backpropagation. Specifically, we design a Dual-Path Gradient Compensator (DPGC) that constructs a parallel full-precision auxiliary branch for each binarized layer, decoupling gradient flow via output decomposition during backpropagation. DPGC enables bias-reduced gradient estimation by leveraging the full-precision branch to estimate components beyond STE's first-order approximation. To further enhance training stability, we introduce an Adaptive Gradient Scaler (AGS) based on an optimal scale factor to dynamically balance inter-branch gradient contributions via norm-based scaling. Experiments on image classification, object detection, and language understanding tasks demonstrate that SURGE performs best over state-of-the-art methods.

---

[*]Equal contribution. [†]Project lead. [‡]Corresponding author.
[1]National College for Excellent Engineers, Beihang University, China [2]School of Artificial Intelligence, Beihang University, China [3]State Key Laboratory of Media Convergence and Communication, Communication University of China, China [4]School of Electronic and Information Engineering, Beihang University, China [5]King Abdullah University of Science and Technology, Saudi Arabia [6]Huawei Noah's Ark Lab, China [7]Hangzhou Innovation Research Institute, Beihang University, China. Correspondence to: Linlin Yang <lyang@cuc.edu.cn>.

*Proceedings of the 43$^{rd}$ International Conference on Machine Learning*, Seoul, South Korea. PMLR 306, 2026. Copyright 2026 by the author(s).

## 1. Introduction

Deep neural networks (DNNs) have achieved remarkable success across various domains (He et al., 2016; Vaswani, 2017), with model parameters scaling from millions to billions in state-of-the-art architectures (Brown et al., 2020; Yang et al., 2024). However, their escalating computational complexity and memory requirements pose significant challenges for deployment in resource-limited scenarios. To address this challenge, numerous model compression techniques have been developed to enhance deployment efficiency (He & Xiao, 2023; Hinton et al., 2014; Liu et al., 2025; Yu et al., 2017), each offering distinct trade-offs among compression ratio, inference speedup, and accuracy retention. Different from structural compression methods (*e.g.*, pruning), quantization (Esser et al., 2019; Hubara et al., 2021; Wang et al., 2022; Xu et al., 2023) achieves compression through bit-width reduction without modifying the network architecture. The reduced bit-width representation significantly decreases storage requirements while enabling computational acceleration via low-precision operations.

As an extreme form of quantization, binarization (Courbariaux et al., 2015; 2016; Gong et al., 2019; Xu et al., 2021b; 2022a) represents weights and activations with 1-bit values, theoretically enabling $32\times$ memory reduction and $58\times$ computational acceleration compared to full-precision networks (Rastegari et al., 2016). These efficiency advantages of binarization make it especially practical for edge computing devices with severely limited computational resources, and its effectiveness has been proven in diverse tasks, such as classification (Xu et al., 2021c), object detection (Xu et al., 2022b), and natural language understanding (Qin et al., 2022).

Despite considerable advances, there remains a non-negligible performance gap between binary neural networks (BNNs) and their full-precision counterparts (Rastegari et al., 2016). This discrepancy primarily stems from the substantial representation divergence between binary and continuous-valued weights and activations. Specifically, the training of BNNs incorporates quantization of real-valued tensors with deterministic or stochastic binarization operations (Courbariaux et al., 2016). However, the non-

*Figure 1.* (a-b) Activation gradient patterns without/with SURGE (left/right); (c) Gradient distribution comparison; (d) Cumulative probability of gradients. STE provides a first-order approximation for the `sign` function's gradient and clips out-of-range activation gradients, while SURGE compensates them with a Dual-Path Gradient Compensator (a-b). SURGE also right-shifts gradient distributions of activations (c-d), validating its effectiveness in rectifying STE-induced mismatch.

differentiable nature and vanishing gradients of binarization operations introduce significant challenges in backpropagation.

To solve the training problem, the Straight-Through Estimator (STE) (Bengio et al., 2013) provides an effective gradient approximation method for binarization operations. Specifically, STE directly substitutes the gradients of binarization operations (*e.g.*, `sign` function) with the derivative of the `Identity` function during backpropagation, thereby enabling stable parameter optimization. Despite its prevalent application in training BNNs and low-bit networks, STE suffers from several inherent limitations that remain to be addressed. On the one hand, since the `sign` function's gradient vanishes everywhere except at zero, employing a fixed-value gradient approximation inevitably introduces estimation bias and optimization instability (Qin et al., 2020). To reduce the gradient error of STE, subsequent approaches predominantly rely on heuristic quantizer designs (Liu et al., 2019; Gong et al., 2019), such as piecewise polynomial functions (Liu et al., 2018b) and SignSwish activation functions (Darabi et al., 2018), which cannot guarantee finding the optimal gradient approximation.

On the other hand, during the backpropagation of STE, the gradient clipping is adopted to only preserve the gradient for inputs within the vicinity of zero (typically $[-1, 1]$), which empirically improves model accuracy (Courbariaux et al., 2016). However, applying fixed-range gradient clipping is suboptimal for binarized representations, particularly for activation quantization, since the gradient information is discarded for values outside the clipping range (Qin et al., 2020). Existing binarization methods largely overlook the impact of gradient clipping range, as only a few studies propose handcrafted asymptotic functions to gradually approximate the hard binarization function (Gong et al., 2019; Qin et al., 2020). Consequently, merely employing STE and improved estimators (Rastegari et al., 2016; Gong et al., 2019; Xu et al., 2022a; Jin et al., 2025) fails to obtain accu-

rate gradient approximation for binarization operations, as non-negligible gradient mismatch (Qin et al., 2020) accumulates in the backward pass, necessitating explicit gradient rectification.

This paper proposes SURrogate GradiEnt Adaptation (SURGE), a novel learnable gradient compensation strategy that addresses gradient mismatch through auxiliary backpropagation. While STE or improved estimators provides surrogate gradients for binarization operations, SURGE offers enhanced gradient adaptation for binary neural networks. Specifically, we design a Dual-Path Gradient Compensator (DPGC), which constructs a parallel full-precision parameterized branch (noted as auxiliary branch) for each binarized layer (noted as main branch). In particular, DPGC decomposes each layer's output into contributions from the main branch and auxiliary branch, thus decoupling the gradient flow into two parts during backpropagation. Therefore, DPGC ensures that the auxiliary branch only affects the backward gradient while preserving the original layer outputs during the forward pass. Compared with the binary branch, the full-precision branch can provide less biased gradients (Stock et al., 2021) that compensate for STE's first-order approximation (Liu et al., 2023) error by learning higher-order terms. As shown in Figure 1, **(a)** STE's fixed clipping *zeros vast area* of activation gradients; **(b)** with SURGE, the auxiliary branch injects compensation gradients while keeping the forward output unchanged, visibly recovering the clipped regions. Aggregated statistics in **(c)**–**(d)** show a right-shifted gradient distribution and heavier tails in the cumulative curves, indicating that SURGE restores informative gradients beyond STE's first-order surrogate.

Moreover, large-magnitude gradients from the auxiliary path may adversely affect the convergence of the main branch. To address this problem, we propose an Adaptive Gradient Scaler (AGS) that dynamically balances inter-branch gradient contributions via norm-based scaling, thereby ensuring

stable and effective compensation. To validate the effectiveness of SURGE, we conduct comprehensive comparative experiments on two image classification benchmarks, one object detection benchmark, one suite of language understanding benchmark, and our proposed method achieves best performance over state-of-the-art. In summary, the main contributions of this work are as follows:

- We propose SURrogate GradiEnt Adaptation (SURGE), a novel gradient compensation framework employing a Dual-Path Gradient Compensator to address gradient mismatch. Our method does not modify the forward-pass output and introduces no additional overhead at inference.

- We introduce an Adaptive Gradient Scaler (AGS) that dynamically equilibrates gradient contributions from binary and auxiliary branches based on theoretically derived optimal scaling factor.

- Extensive experiments demonstrate that SURGE achieves state-of-the-art performance across four standard benchmarks for BNN training. Specifically, a SURGE-trained binarized ResNet-18 attains 62.0% top-1 accuracy on ImageNet with one-stage training, surpassing previous SOTA methods by significant margins (*e.g.*, +1.0%, and +3.9% top-1 accuracy improvements over ReCU and IR-Net, respectively, on ImageNet).

## 2. Related Work

### 2.1. Gradient Approximation

Gradient approximation serves as a cornerstone for training neural networks with non-differentiable operators, addressing challenges in discrete sampling (Sutton et al., 1999; Schulman et al., 2015; Athalye et al., 2018; Rezende et al., 2014), architecture search (Xie et al., 2018; Liu et al., 2018a; Cai et al., 2018), and especially quantization (Esser et al., 2020; Gong et al., 2019; Liu et al., 2018b; 2020; Xu et al., 2022a). A popular family of gradient estimators is the Straight-Through Estimator (STE), which directly propagates gradients through non-differentiable functions. The idea of Straight-Through originates from the perceptron+ algorithm (Rosenblatt, 1957), which leverages a modified chain rule and utilizes the Identity function as the proxy of the original derivative of a binary output function. (Bengio et al., 2013) improves this method by using non-linear functions like sigmoid, and (Jang et al., 2016) further incorporates the Gumbel reparameterization, reparameterizes discrete variables via temperature-annealed continuous relaxation, enabling low-variance gradient estimation for categorical sampling. In the field of quantization, DSQ (Gong et al., 2019) employs parameterized sigmoid functions to progres-

sively approximate the gradients of the non-differentiable quantization function, while LSQ (Esser et al., 2020) introduced scaling factors for end-to-end gradient propagation, advancing low-bit quantization. BONN (Zhao et al., 2022) integrates Bayesian optimization to guide differentiable binarization policies, and FDA-BNN (Xu et al., 2021b) converts the `sign` function into the frequency domain to mitigate the gradient mismatch.

### 2.2. Binary Neural Network

Pioneering works in binary neural networks focused either on binarization architecture design (Liu et al., 2018b; Xu et al., 2021b; Liu et al., 2020; Bulat et al., 2020; Yang et al., 2020) or training strategies (Courbariaux et al., 2015; Rastegari et al., 2016; Qin et al., 2020; Xu et al., 2021c; 2022a). In terms of architecture design, Bi-Real Net (Liu et al., 2018b) enhances skip connections, and FDA-BNN (Xu et al., 2021b) introduces differentiable binarization units in the frequency domain. Moreover, ReActNet (Liu et al., 2020) substitutes the `sign` function and PReLU (He et al., 2015) with RSign and RPReLU based on learnable thresholds. Approaches like BATS (Bulat et al., 2020) and SLB (Yang et al., 2020) combine BNNs with neural architecture search. In terms of training strategies, BinaryConnect (Courbariaux et al., 2015) and XNOR-Net (Rastegari et al., 2016) use the `sign` function with gradient approximation, but they cause severe information loss in forward propagation. Later, training strategies were innovated. IR-Net (Qin et al., 2020) and ReCU (Xu et al., 2021c) use progressive quantization and feature distribution alignment, but they still face gradient mismatch in deep networks. RBONN (Xu et al., 2022a) introduces a recurrent bilinear optimization for BNNs.

Unlike prior work, our work is the first attempt to employ a Dual-Path Gradient Compensator to correct gradient mismatch in STE-based binarized networks, coupled with an Adaptive Gradient Scaler to equilibrate the gradient contribution between binary and auxiliary branches dynamically.

## 3. Preliminaries

Consider a neural layer with weight vector $W \in \mathbb{R}^d$ and input vector $x \in \mathbb{R}^d$. The main operation in deep neural networks is expressed as:

$$f(x; W) = W^\top x. \tag{1}$$

In binary neural networks (BNNs), we quantize $W$ and $x$ to $\{-1, +1\}^d$, thus using the efficient XNOR and Bitcount operations to replace real-valued operations. Let $\mathbf{B}_W \in \{-1, +1\}^d$ and $\mathbf{B}_x \in \{-1, +1\}^d$ denote the binarized counterparts. Network binarization aims to represent the floating-point weights and/or activations with 1 bit. In general, the quantization can be formulated as:

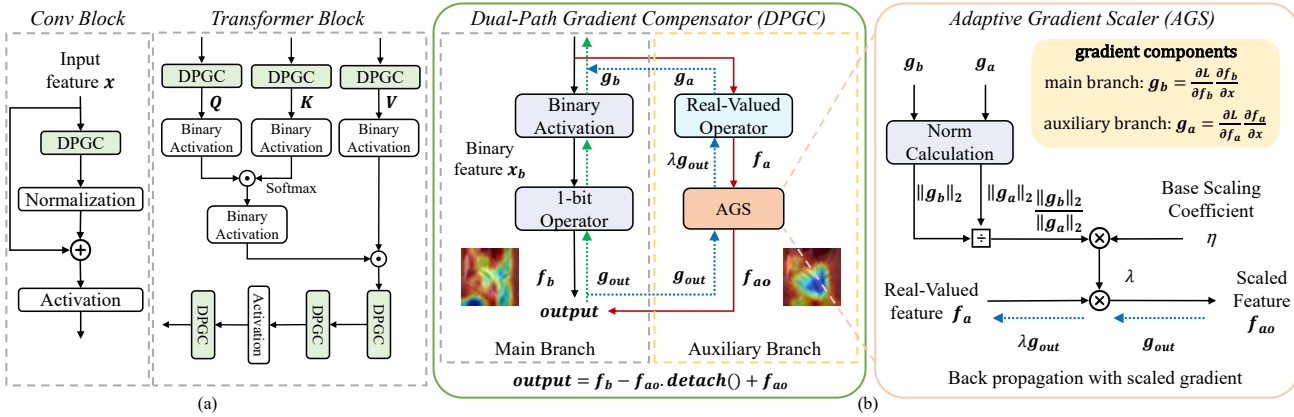

*Figure 2.* Overall architecture of SURGE. **(a)** Integration into common backbones (left: convolution block; right: transformer block). For visual clarity, residual connections are omitted in the Transformer block. **(b)** Component details. DPGC constructs a parallel full-precision parameterized branch (auxiliary branch, shown with red arrows for forward pass and blue arrows for backpropagation) for each binarized layer (main branch, represented by black arrows in forward pass and green arrows for backpropagation). This ensures identical output to standard BNNs while providing less biased gradients for compensation. AGS takes gradients from both branches as input (visualized through corresponding colored arrows) and dynamically balances inter-branch gradient contributions via norm-based scaling. SURGE is architecture-agnostic and applies to arbitrary binarized linear operators.

$Q_x(x) = \alpha_x \mathbf{B}_x, Q_W(W) = \alpha_W \mathbf{B}_W$, where $\alpha$. denotes scalars for binary values including $\alpha_w$ for weights and $\alpha_x$ for inputs. And we usually use `sign` function to binarize $W$ and $x$: $\mathbf{B}_x = \text{sign}(x), \mathbf{B}_W = \text{sign}(W)$. Following (Rastegari et al., 2016), the binary operation is formulated as:

$$f_b(x; \mathbf{B}_W) = Q_W(W)^\top Q_x(x) = \alpha_W \alpha_x \cdot (\mathbf{B}_W \odot \mathbf{B}_x), \tag{2}$$

where $\odot$ denotes the inner product for vectors with bitwise operations XNOR and Bitcount.

In backpropagation, the derivative of the `sign` function is zero almost everywhere, which makes it incompatible with backpropagation, since exact gradients for the original values before binarization would be zeroed. Thus, Straight-Through Estimator (STE) (Bengio et al., 2013) is generally used to train BNNs, which propagates the gradient through `Identity` function. Regarding the gradient of the loss $L$ *w.r.t.* $W$, it is approximated as

$$\frac{\partial L}{\partial W} = \frac{\partial L}{\partial \mathbf{B}_W} \cdot \frac{\partial \mathbf{B}_W}{\partial W} \approx \frac{\partial L}{\partial \mathbf{B}_W}. \tag{3}$$

As for the gradient *w.r.t.* the activations, it can be formulated as

$$\frac{\partial L}{\partial x} = \frac{\partial L}{\partial \mathbf{B}_x} \cdot \frac{\partial \mathbf{B}_x}{\partial x} \approx \frac{\partial L}{\partial \mathbf{B}_x} \cdot \mathbf{1}_{\{|x| \leq 1\}}, \tag{4}$$

where $\mathbf{1}_{\{|x| \leq 1\}}$ is the indicator function that equals 1 when $|x| \leq 1$ and 0 otherwise. This expression corresponds to STE's first-order approximation for the sign function's gradient.

## 4. Methodology

In this section, we describe SURGE in detail. We first introduce the Dual-Path Gradient Compensator (DPGC) module to address the gradient mismatch in STE-based training (Sec. 4.1), then present the Adaptive Gradient Scaler (AGS) for stable optimization (Sec. 4.2). The complete training paradigm integrates these components while preserving standard BNN inference.

### 4.1. Dual-Path Gradient Compensator (DPGC)

To handle the intrinsic gradient mismatch in STE (Qin et al., 2020), we propose a layer-wise dual-path architecture that preserves original forward computations while introducing auxiliary gradient pathways. As shown in Figure 2, DPGC constructs a parallel full-precision parameterized branch (noted as auxiliary branch) including a full-precision operator (*e.g.*, convolution, linear, attention projection) with identical dimensions (*e.g.*, kernel size, dimension) to the main branch, augmented with an Adaptive Gradient Scaler (AGS) module (Section 4.2) for each binarized layer (noted as main branch). DPGC decomposes each layer's output into contributions from both the main branch (black arrow) and auxiliary branch (red arrow), thus decoupling the gradient flow into two parts during backpropagation (Eq. 6) (green arrow for main branch, blue arrow for auxiliary branch). Given $f_b(x) = Q_W(W_b)^\top Q_x(x)$ as the binarized computation, $f_a(x) = W_a^\top x$ as the full-precision computation, $W_a, W_b$ as the weight parameters for auxiliary branch (full-precision branch), main branch (binary branch), respectively, the combined output is:

$$\text{output} = \underbrace{f_b(x; W_b)}_{\text{Binary output}} - \underbrace{f_{ao}(x; W_a) \downarrow}_{\text{Detached compensator}} + \underbrace{f_{ao}(x; W_a)}_{\text{Active compensator}} \,,$$

$$(5)$$

where $f_{ao}(x) = \lambda f_a(x)$ is the scaled full-precision computation, $\lambda$ is the scale factor, $\downarrow$ is the gradient stop operator. The design ensures identical outputs to standard BNNs, while gradients flow through both pathways, thus providing less biased gradient estimates (Stock et al., 2021) while preserving STE gradients. Upon completion of training, the auxiliary branch can be discarded, introducing no additional computational overhead during inference. Backpropagation aggregates gradients from both paths:

$$\frac{\partial \mathcal{L}}{\partial x} = \underbrace{\frac{\partial \mathcal{L}}{\partial f_b} \frac{\partial f_b}{\partial x}\Big|_{\text{STE}}}_{\text{Binary gradients } g_b} + \lambda \underbrace{\frac{\partial \mathcal{L}}{\partial f_a} \frac{\partial f_a}{\partial x}}_{\text{Compensator gradients } g_a} \,. \quad (6)$$

Here, $\frac{\partial f_b}{\partial x}\big|_{\text{STE}}$ serves as the STE surrogate for the theoretically intractable binary-branch derivative (Liu et al., 2023). The term $g_b$ is thereby a first-order approximation, while the full-precision auxiliary branch yields $g_a$ to capture higher-order terms and recover gradients removed by clipping.

### 4.2. Adaptive Gradient Scaler (AGS)

The raw combination of $g_b$ and $g_a$ risks unstable training due to varying magnitude ratios between paths, and the large-magnitude gradients from the auxiliary path may adversely affect the convergence of the main branch. We address this through a novel mechanism that dynamically balances inter-branch gradient contributions with norm-based adaptive scaling factor $\lambda_{\text{AGS}}$, thereby ensuring stable and effective compensation:

$$\frac{\partial \mathcal{L}}{\partial x} = g_b + \lambda_{\text{AGS}} \cdot g_a, \quad \lambda_{\text{AGS}} := \eta \frac{\|g_b\|_2}{\|g_a\|_2 + \epsilon}, \quad (7)$$

where $\eta$ is the base scaling coefficient, $\epsilon = 10^{-8}$ is the numerical stabilizer, and $\lambda_{\text{AGS}}$ is a practical plug-in approximation of the theoretical optimum (Theorem 5.3, Corollary 5.4). This dynamic scaling preserves the directional consistency of the primary binary gradient $g_b$ while allowing auxiliary gradients $g_a$ to provide magnitude-aware compensation. Such design guarantees that the STE-based gradients dominate the parameter update process, while the auxiliary path serves as an adaptive compensator that injects higher-order gradient information without destabilizing the primary learning dynamics. In practice, the scale factor derived from gradient computation in the current iteration is used in the subsequent AGS step for adaptive parameter adjustment to

optimize computational efficiency. The complete training procedure is summarized in Appendix A.

## 5. Theoretical Analysis

This section formally establishes the theoretical foundation of gradient compensation in dual-path architectures. We begin by formulating the gradient propagation mechanism under our proposed compensation framework, followed by introducing moment-based notation for gradient statistics (Definition 5.1) and a moment model that captures the bias/noise structure of the two gradient components (Assumption 5.2). We then derive the theoretically optimal scaling factor for gradient compensation (Theorem 5.3) and obtain a practical norm-ratio approximation that directly motivates the AGS update rule (Corollary 5.4).

Let $\mathcal{X}$ denote the input space and $W = (W_b, W_a) \in \mathbb{R}^{2d}$ represent the binarized and full-precision weights. The forward propagation becomes:

$$f(x; W) = $$
$$\underbrace{Q_W(W_b)^\top Q_x(x)}_{\text{Binary path}} + \lambda \left( \underbrace{W_a^\top x}_{\text{Compensator path}} - \underbrace{W_a^\top x \downarrow}_{\text{Detached path}} \right),$$

$$(8)$$

where $\lambda$ follows the adaptive scaling in Section 4.2. Let *Approx* denote a kind of STE-based gradient approximation (*e.g.*, STE), the composite gradient combines:

$$\frac{\partial \mathcal{L}}{\partial x} = \underbrace{\frac{\partial \mathcal{L}}{\partial f_b} \frac{\partial f_b}{\partial x}\Big|_{Approx}}_{g_b} + \underbrace{\frac{\partial \mathcal{L}}{\partial f_{ao}} \frac{\partial f_{ao}}{\partial x}}_{\lambda g_a} \,. \quad (9)$$

**Definition 5.1** (Notation for gradient statistics). Let $\mu_b := \mathbb{E}[g_b]$, $\mu_a := \mathbb{E}[g_a]$. Define the approximation-induced bias vector of the baseline surrogate as $\delta_b := g^* - \mu_b$.

**Assumption 5.2** (Moment model for gradient components). Let $\mu_b := \mathbb{E}[g_b]$ and $\mu_a := \mathbb{E}[g_a]$. There exists an ideal (unobserved) reference gradient $g^*$ such that

$$\mu_b = g^* - \delta_b, \qquad \|\delta_b\|_2 \leq C\sqrt{d},$$

and $g_b, g_a$ have finite second moments.

The definition of $g^*$ and the noise structure assumptions are detailed in Appendix B.1.

**Theorem 5.3** (Optimal scaling factor). *Let* $\tilde{g}(\lambda) := g_b + \lambda g_a$. *Under Assumption 5.2, assume* $\mu_a := \mathbb{E}[g_a]$ *exists and* $\text{Var}(g_a)$ *has finite trace. Assume additionally that the mini-batch noises in* $g_b$ *and* $g_a$ *are uncorrelated in the dot-product sense:*

$$\mathbb{E}\big[(g_b - \mathbb{E}[g_b])^\top (g_a - \mathbb{E}[g_a])\big] = 0.$$

Then any minimizer of $\mathbb{E}\|\tilde{g}(\lambda) - g^*\|_2^2$ is

$$\lambda^* = \frac{\langle \delta_b, \, \mu_a \rangle}{\|\mu_a\|_2^2 + \mathrm{tr}(\mathrm{Var}(g_a))}.$$

If $\|\mu_a\|_2^2 + \mathrm{tr}(\mathrm{Var}(g_a)) > 0$, this minimizer is unique. In particular, under the isotropic noise model $\mathrm{Var}(g_a) = \sigma_a^2 I_d$,

$$\lambda^* = \frac{\langle \delta_b, \, \mu_a \rangle}{\|\mu_a\|_2^2 + d\sigma_a^2}.$$

**Corollary 5.4** (Practical norm-ratio approximation). *In addition, suppose during the main training phase (after a short transient): (i) the alignment* $\cos\theta := \frac{\langle \delta_b, \mu_a \rangle}{\|\delta_b\|_2 \|\mu_a\|_2} \approx c_\theta$ *is approximately stable, (ii) the* relative bias ratio $\beta := \frac{\|\delta_b\|_2}{\|\mu_b\|_2}$ *with* $\mu_b := \mathbb{E}[g_b]$ *is bounded and slowly varying, so* $\beta \approx \kappa$, *and (iii) the noise ratio* $\rho := \frac{d\sigma_a^2}{\|\mu_a\|_2^2}$ *is approximately stable. Then*

$$\lambda^* \approx \eta \frac{\|\mu_b\|_2}{\|\mu_a\|_2}, \qquad \eta := \frac{\kappa c_\theta}{1 + \rho}.$$

*Replacing population quantities by mini-batch estimates and adding a numerical stabilizer* $\epsilon > 0$ *yields the AGS rule*

$$\lambda_{\mathrm{AGS}} := \eta \frac{\|g_b\|_2}{\|g_a\|_2 + \epsilon}.$$

The proof is detailed in Appendix B.2. We now derive the practical expression $\lambda_{\mathrm{AGS}}$ for the optimal scaling factor $\lambda^*$, which is norm-based and adaptive, adopted in our AGS module (Sec. 4.2). This analysis establishes a principled gradient scaling factor that improves the resulting gradient update and alleviates the gradient mismatch.

# 6. Experiments

## 6.1. Datasets and Implementation Details

**Datasets.** We evaluate on two standard image classification benchmarks, one object detection benchmark, and one suite of language understanding tasks to demonstrate the effectiveness: CIFAR-10 (Krizhevsky et al., 2009), ImageNet-1K (Russakovsky et al., 2015), PASCAL VOC (Everingham et al., 2010), and GLUE (Wang et al., 2018). More details of datasets, data augmentation, and evaluating metrics are provided in Appendix D.

**Implementation Details.** On CIFAR-10, we evaluate our method with ResNet-18/20 (He et al., 2016) and VGG-Small (Simonyan & Zisserman, 2014). On PASCAL VOC, we binarize Faster-RCNN (Ren et al., 2016) with a ResNet-18 backbone (with minor structural modifications shared by FP/BNN). On GLUE, we evaluate our method with BERT-base (Devlin et al., 2019). More training details are provided in Appendix E.

*Table 1.* Performance comparison with the state-of-the-arts on CIFAR-10. W/A denotes the bit length of the weights and activations.

| Network | Method | W/A | Top-1 |
|---|---|---|---|
| ResNet-18 | Real-Valued | 32/32 | 94.8% |
| | RAD | 1/1 | 90.5% |
| | IR-Net | 1/1 | 91.5% |
| | RBNN | 1/1 | 92.2% |
| | ReCU | 1/1 | 92.8% |
| | **SURGE (Ours)** | 1/1 | **93.1%** |
| ResNet-20 | Real-Valued | 32/32 | 92.1% |
| | DoReFa | 1/1 | 79.3% |
| | DSQ | 1/1 | 84.1% |
| | SLB | 1/1 | 85.5% |
| | IR-Net | 1/1 | 86.5% |
| | ReCU | 1/1 | 87.4% |
| | **SURGE (Ours)** | 1/1 | **88.0%** |
| VGG-Small | Real-Valued | 32/32 | 94.1% |
| | XNOR-Net | 1/1 | 89.8% |
| | DoReFa | 1/1 | 90.2% |
| | IR-Net | 1/1 | 90.4% |
| | RBNN | 1/1 | 91.3% |
| | DSQ | 1/1 | 91.7% |
| | SLB | 1/1 | 92.0% |
| | ReCU | 1/1 | 92.2% |
| | **SURGE (Ours)** | 1/1 | **92.5%** |

## 6.2. Image Classification

**CIFAR-10.** We first show the experimental results on CIFAR-10 with ResNet-18, ResNet-20, VGG-Small backbone in Table 1. Specifically, we compare SURGE with state-of-the-art methods include RAD (Ding et al., 2019), IR-Net (Qin et al., 2020), RBNN (Lin et al., 2020), ReCU (Xu et al., 2021c), DoReFa (Zhou et al., 2016), DSQ (Gong et al., 2019), SLB (Yang et al., 2020), IR-Net (Qin et al., 2020), and XNOR-Net (Rastegari et al., 2016). We can see that SURGE outperforms all other methods in all backbones. Compared to recent ReCU, SURGE obtains a 0.3% performance increase with ResNet-18, a 0.6% performance increase with ResNet-20, and a 0.3% performance increase with VGG-Small.

**One-Stage Training on ImageNet.** Table 2 displays the performance comparison in binarizing ResNet-18 with one-stage training on ImageNet. We compare SURGE with DoReFa (Zhou et al., 2016), TBN (Wan et al., 2018), BNN (Courbariaux et al., 2016), XNOR-Net (Rastegari et al., 2016), Bi-Real Net (Liu et al., 2018b), IR-Net (Qin et al., 2020), BONN (Zhao et al., 2022), RBNN (Lin et al., 2020), RBONN (Xu et al., 2022a). We can see that SURGE is leading in both the top-1 and top-5 accuracies. Specifically, SURGE outperforms RBONN by 0.6% in top-1 accuracy, achieving the best performance.

*Table 2.* A performance comparison with SOTAs on ImageNet with one-stage training. W/A denotes the bit length of weights and activations. We report the Top-1 (%) and Top-5 (%) accuracy performances.

| Network | Method | W/A | OPs ($\times 10^8$) | Top-1 | Top-5 |
|---------|--------|-----|---------------------|-------|-------|
| ResNet-18 | Real-valued | 32/32 | 18.19 | 69.6 | 89.2 |
| | DoReFa | 1/4 | 2.44 | 59.2 | 81.5 |
| | TBN | 1/2 | 1.81 | 55.6 | 79.0 |
| | BNN | 1/1 | 1.63 | 42.2 | 67.1 |
| | XNOR-Net | | | 51.2 | 73.2 |
| | Bi-Real Net | | | 56.4 | 79.5 |
| | IR-Net | | | 58.1 | 80.0 |
| | BONN | | | 59.3 | 81.6 |
| | RBNN | | | 59.6 | 81.6 |
| | ReCU | | | 61.0 | 82.6 |
| | RBONN | | | 61.4 | 83.5 |
| | **SURGE (Ours)** | | | **62.0** | **83.7** |

*Table 3.* A performance comparison with SOTAs on ImageNet with two-stage training. W/A denotes the bit length of weights and activations. We report the Top-1 (%) and Top-5 (%) accuracy performances. * denotes the result is from the official checkpoint.

| Network | Method | W/A | OPs ($\times 10^8$) | Top-1 | Top-5 |
|---------|--------|-----|---------------------|-------|-------|
| ResNet-18 | Real-valued | 32/32 | 18.19 | 69.6 | 89.2 |
| | ReActNet | 1/1 | 1.63 | 65.9 | - |
| | ReCU | | | 66.4 | 86.5 |
| | RBONN* | | | 66.5 | **86.7** |
| | **SURGE (Ours)** | | | **66.7** | **86.7** |

**Two-Stage Training on ImageNet.** Table 3 displays the performance comparison in binarizing ResNet-18 with two-stage training on ImageNet. We compare SURGE with ReActNet (Liu et al., 2020), ReCU (Xu et al., 2021c), and RBONN (Xu et al., 2022a). Results show that SURGE outperforms all other methods in top-1 accuracy. Specifically, SURGE obtains a 0.2% performance increase in top-1 over RBONN. Our SURGE demonstrates superior overall performance compared to all existing approaches.

### 6.3. Object Detection

On the PASCAL VOC dataset, we compare the proposed SURGE against existing state-of-the-art binarized detection methods, such as ReActNet (Liu et al., 2020), LWS-Det (Xu et al., 2021a), and IDa-Det (Xu et al., 2022b), on the Faster-RCNN framework for object detection. The detection result of multi-bit quantized networks DoReFa-Net (Zhou et al., 2016) is also reported. As shown in Table 4, compared with the prior state-of-the-art IDa-Det, our method gains 0.5% performance increase, with the same FLOPs and memory usage. Compared with the raw real-valued detectors, SURGE surpasses raw real-valued Faster-RCNN with ResNet-18 backbone (77.0% *v.s.* 76.4%) by apparent computation acceleration and storage savings by $5.21\times$ and $6.80\times$.

### 6.4. Language Understanding

On the GLUE dataset, we compare SURGE against existing state-of-the-art methods, such as BinaryBERT (Bai et al., 2020), BiBERT (Qin et al., 2022), and BiT (Liu et al., 2022), on BERT. We can see that SURGE outperforms all other methods. Specifically, SURGE obtains a 1.4% performance increase compared to BiT, and outperforms BiBERT by 8.9%, achieving the best performance.

### 6.5. Ablation Study

**Ablation on Components.** We ablate each component on CIFAR-10 using ResNet20. As shown in Table 6a, the baseline achieves 87.4% accuracy. Introducing the **Dual-Path Gradient Compensator (DPGC)** alone improves performance by +0.4%, validating its capability to balance gradient conflicts. Subsequent integration of the **Adaptive Gradient Scaler (AGS)** adds another +0.2%, demonstrating that AGS effectively modulates gradient magnitudes without disrupting DPGC's compensation. The hierarchical gains confirm that both mechanisms address distinct aspects of gradient optimization.

**Ablation on Parameter $\eta$.** As shown in Figure 3, the performance degradation of fixed scaling ($scale > 0.05$, max -17.7%) highlights the necessity of dynamic adaptation, while our adaptive scaling achieves peak accuracy (87.95%) at $\eta = 0.01$. The result confirms that our theory-

*Table 4.* Performance comparison of different methods in Faster-RCNN framework with input resolution set to 1000×600. [†] denotes that the result is from our re-implementation.

| Framework | Backbone | Method | W/A | Memory Usage (MB) | OPs ($\times 10^9$) | mAP |
|---|---|---|---|---|---|---|
| Faster-RCNN | ResNet-18 | Real-valued | 32/32 | 112.88 | 96.40 | 78.8 |
| | | DoReFa-Net | 4/4 | 21.59 | 27.15 | 73.3 |
| | | ReActNet | 1/1 | 16.61 | 18.49 | 69.6 |
| | | LWS-Det | | | | 73.2 |
| | | IDa-Det[†] | | | | 76.5 |
| | | **SURGE (Ours)** | | | | **77.0** |

*Table 5.* Performance comparison of BERT quantization on the GLUE dev set. FP is short for full precision. [†] denotes our re-implementation without multi-distillation techniques for fair comparison.

| Quant | Size (MB) | FLOPs (G) | MNLI$_{m/mm}$ | QQP | QNLI | SST-2 | CoLA | STS-B | MRPC | RTE | Avg. |
|---|---|---|---|---|---|---|---|---|---|---|---|
| BERT (FP) | 418 | 22.5 | 84.9/85.5 | 91.4 | 92.1 | 93.2 | 59.7 | 90.1 | 86.3 | 72.2 | 83.9 |
| BinaryBERT | 16.5 | 0.4 | 35.6/35.3 | 66.2 | 51.5 | 53.2 | 0 | 6.1 | 68.3 | 52.7 | 41.0 |
| BiBERT | 13.4 | 0.4 | 66.1/67.5 | 84.8 | 72.6 | 88.7 | 25.4 | 33.6 | 72.5 | 57.4 | 63.2 |
| BiT[†] | 13.4 | 0.4 | 77.0/77.5 | 85.4 | 85.5 | 87.8 | 23.6 | 68.0 | 79.4 | 58.1 | 70.6 |
| **SURGE(Ours)** | 13.4 | 0.4 | 77.3/77.5 | 87.1 | 86.2 | 88.6 | 24.1 | 71.7 | 80.6 | 60.6 | **72.0** |

*Table 6.* Ablation Study on CIFAR-10 with ResNet20.

*(a)* Ablation on components

| Method | Accuracy (%) |
|---|---|
| Baseline | 87.4 |
| + DPGC | 87.8 |
| + DPGC + AGS | **88.0** |

*(b)* Ablation on gradient compensation scope

| Method | Scope | Accuracy (%) |
|---|---|---|
| Baseline | / | 87.4 |
| SURGE | clipped gradients | 87.7 |
| | unclipped gradients | 87.6 |
| | all gradients | **88.0** |

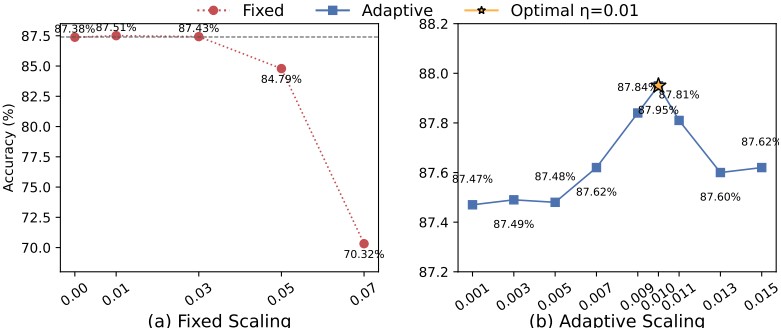

*Figure 3.* Ablation study on parameter scaling strategies. (a) is fixed scaling with constant factors across training iterations. (b) is adaptive scaling via parameter $\eta$ that dynamically adjusts the compensation strength (Eq. 7).

driven design (Theorem 5.3) successfully balances gradient compensation and training stability.

**Ablation on Gradient Compensation Scope of DPGC.** We ablate the gradient compensation scope on CIFAR-10 using ResNet20. As detailed in Table 6b, compensating *only* gradients outside STE's clipping range ($|x| > 1$) yields 87.7% accuracy (0.3% improvement over baseline), while compensating *solely* within-range gradients ($|x| \leq 1$) achieves 87.6% (0.2% improvement). This verifies that both clipped and preserved gradient components contribute to

parameter optimization. When jointly compensating *all* activation gradients through SURGE's adaptive integration, accuracy rises to 88.0% (0.6% improvement). This confirms that SURGE's design overcomes fixed-range clipping limitations in STE, enabling comprehensive gradient utilization.

## 7. Conclusion

This paper proposes a novel gradient compensation strategy that mitigates the STE-induced gradient mismatch through an auxiliary backpropagation. The proposed Dual-Path Gra-

dient Compensator (DPGC) utilizes a dual-path architecture that ensures identical output to standard BNNs while providing less biased gradients for compensation. And the Adaptive Gradient Scaler (AGS) dynamically balances inter-branch gradient contributions via norm-based scaling. SURGE obtains the best performance over existing methods through main benchmarks in image classification, object detection, and language understanding tasks.

## Acknowledgements

The work was supported by the National Key Research and Development Program of China (No.2023YFC3306401) and Beijing Natural Science Foundation L244043 and L2607011. This research was also supported by National Natural Science Foundation of China 623B2016 and 62576018, and Zhejiang Provincial Natural Science Foundation of China under Grant No. LD24F020007.

## Impact Statement

This paper proposes a training-time gradient compensation framework for Binary Neural Networks (BNNs) to improve optimization stability and accuracy under extreme quantization. The primary expected impact is enabling more efficient deployment of neural models on resource-constrained devices through reduced memory footprint and computation, which may lower energy consumption and operational cost for practical applications.

We do not anticipate direct safety or security risks introduced by the proposed method beyond those already associated with deploying machine learning models in real-world settings. The method does not introduce new data collection, does not require sensitive information, and does not change the functional scope of the underlying models; it mainly affects the training dynamics and can be removed at inference time. As with other model compression techniques, improved efficiency could facilitate wider deployment, and responsible use should follow standard best practices for dataset governance, evaluation, and monitoring in downstream applications.

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

# APPENDIX

## A. Training Procedure

The complete training procedure is summarized in Algorithm A.

---

**Algorithm A** Layer-wise Training with DPGC & AGS

---

**Require:** Layer input $x^{(l)}$, target $y$, learning rate $\alpha$, base scaling coefficient $\eta$, loss function $\mathcal{F}$
**Ensure:** Trained binarized weights $\{W_b^{(l)}\}_{l=1}^{L}$
1: Initialize layer parameters $W_b^{(l)}, W_a^{(l)}$ {Binary & auxiliary full-precision paths}
2: Initialize $\lambda^{(l,0)} \leftarrow \frac{1}{\sqrt{|W_a^{(l)}|}}$ {Reciprocal sqrt of auxiliary weight cardinality $|W_a^{(l)}|$}
3: **for** iteration $t = 1$ to $T$ **do**
4:     **Forward Propagation (Layer $l$):**
5:     Compute binary path: $f_b^{(l)} \leftarrow W_b^{(l)} \odot \mathrm{Sign}(x^{(l)})$
6:     Compute auxiliary path: $f_a^{(l)} \leftarrow W_a^{(l)} \odot x^{(l)}$
7:     Generate compensator: $f_{ao}^{(l)} \leftarrow \lambda^{(l,t-1)} \odot f_a^{(l)}$ {Previous scaling factor}
8:     Synthesis output: $\mathrm{out}^{(l)} \leftarrow f_b^{(l)} - f_{ao}^{(l)} \downarrow + f_{ao}^{(l)}$ {Forward synthesis via gradient-decoupled decomposition, detach gradient at $\downarrow$}
9:     **Loss Computation:**
10:     Calculate Loss $\mathcal{L}$
11:     **Backward Propagation (Layer $l$):**
12:     Compute main branch gradients:
13:         $g_b \leftarrow \frac{\partial \mathcal{L}}{\partial f_b} \frac{\partial f_b}{\partial x}\Big|_{\mathrm{STE}}, \quad g_{wb}^{(l)} \leftarrow \frac{\partial \mathcal{L}}{\partial W_b^{(l)}}\Big|_{\mathrm{STE}}$
14:     Compute auxiliary branch gradients:
15:         $g_a \leftarrow \frac{\partial \mathcal{L}}{\partial f_a} \frac{\partial f_a}{\partial x}, \quad g_{wa}^{(l)} \leftarrow \lambda^{(l,t-1)} \odot \frac{\partial \mathcal{L}}{\partial W_a^{(l)}}$
16:     Then we have: $\frac{\partial \mathcal{L}}{\partial x^{(l)}} = g_b + \lambda g_a$
17:     **Adaptive Gradient Scaler:**
18:     Calculate norm ratio: $r \leftarrow \|g_b\|_2 / (\|g_a\|_2 + \epsilon)$
19:     Update scaling factor: $\lambda^{(l,t)} \leftarrow \eta \cdot r$
20:     **Parameter Update (Layer $l$):**
21:     $W_b^{(l)} \leftarrow W_b^{(l)} - \alpha \cdot g_{wb}^{(l)}$
22:     $W_a^{(l)} \leftarrow W_a^{(l)} - \alpha \cdot g_{wa}^{(l)}$
23: **end for**

---

## B. Theoretical Foundations and Proofs

### B.1. Assumptions Underlying the Moment Model and Theorem 5.3

In this subsection we make explicit the assumptions underlying Definition 5.1 and Theorem 5.3. Intuitively, since the derivative of `sign` is zero almost everywhere and corresponds to a Dirac delta distribution at the origin in the sense of distributions, it is natural to view $g^*$ as the gradient induced by an "ideal" surrogate that captures this behavior, while practical rules such as STE provide tractable but biased approximations.

**Assumption B.1** (Ideal reference gradient from a surrogate family)**.** Fix a binarization node whose pre-binarization activation is denoted by $x \in \mathbb{R}^d$. Consider a family $\mathcal{S}$ of smooth surrogate functions $s : \mathbb{R} \to \mathbb{R}$ that approximate the non-differentiable `sign` function used in binarization. For each $s \in \mathcal{S}$, let $\ell_s(\cdot; W)$ denote the population loss of the corresponding surrogate network, and define the population risk as a function of this node input:

$$\mathcal{L}_s(x; W) := \mathbb{E}_\xi\big[\ell_s(\xi; W)\big] \quad \text{with the backpropagated gradient taken w.r.t. } x.$$

We assume that there exists a surrogate $s^* \in \mathcal{S}$ that attains the smallest population loss within this family, and define the associated reference ("better") gradient at the current parameter $W$ as

$$g^* := \nabla_x \mathcal{L}_{s^*}(x; W).$$

This $g^*$ is not observable in practice and we never require a closed-form expression for it; it serves as an ideal target that practical surrogate gradients aim to approximate. We assume that $g^*$ has finite second moments.

**Assumption B.2** (Empirical gradients as random vectors). At a fixed parameter $W$, the empirical gradients $g_b, g_a \in \mathbb{R}^d$ obtained from a single mini-batch are modelled as random vectors whose randomness comes from mini-batch sampling, data noise, and the stochastic optimization procedure. All expectations $\mathbb{E}[\cdot]$ and variances $\mathrm{Var}(\cdot)$ in our analysis are taken with respect to this randomness, and the empirical gradients have finite second moments.

**Assumption B.3** (Directional consistency and bounded relative bias). We acknowledge that STE-based gradients may suffer magnitude distortion due to clipping. We assume that the baseline surrogate gradient $g_b$ remains statistically correlated with the descent direction of the ideal reference gradient $g^*$, i.e., $\mathbb{E}[\cos(g_b, g^*)] \geq c > 0$ during the main training phase. Moreover, we assume the *relative bias ratio* $\beta := \|\delta_b\|_2 / \|\mu_b\|_2$ is bounded and varies slowly after a short transient (layer-wise), so it can be treated as approximately constant when deriving practical scaling rules.

**Assumption B.4** (Isotropic, homoscedastic gradient noise). We decompose the empirical gradients as

$$g_b = \mathbb{E}[g_b] + \varepsilon_b, \qquad g_a = \mathbb{E}[g_a] + \varepsilon_a,$$

where the noise terms satisfy $\mathbb{E}[\varepsilon_b] = \mathbb{E}[\varepsilon_a] = 0$. We assume an isotropic, homoscedastic noise model: there exist scalars $\sigma_b^2, \sigma_a^2 \geq 0$ such that

$$\mathrm{Var}(g_b) = \mathbb{E}[\varepsilon_b \varepsilon_b^\top] = \sigma_b^2 I_d, \qquad \mathrm{Var}(g_a) = \mathbb{E}[\varepsilon_a \varepsilon_a^\top] = \sigma_a^2 I_d,$$

where $I_d$ is the $d$-dimensional identity matrix.

## B.2. Proof of Theorem 5.3

*Proof.* Expand the error expectation:

$$\begin{aligned}
\mathbb{E}\big[\|\tilde{g} - g^*\|_2^2\big] &= \mathbb{E}\big[\|g_b + \lambda g_a - g^*\|_2^2\big] \\
&= \|\mathbb{E}[g_b] + \lambda \mathbb{E}[g_a] - g^*\|_2^2 + \mathrm{tr}(\mathrm{Var}(g_b)) + \lambda^2 \mathrm{tr}(\mathrm{Var}(g_a)) \\
&\quad + 2\lambda \, \mathbb{E}\big[(g_b - \mathbb{E}[g_b])^\top (g_a - \mathbb{E}[g_a])\big].
\end{aligned} \tag{B.1}$$

By the dot-product uncorrelated assumption in Theorem 5.3, the last term in (B.1) vanishes. Using $\mathbb{E}[g_b] = g^* - \delta_b$, we obtain

$$\mathbb{E}\big[\|\tilde{g} - g^*\|_2^2\big] = \|\lambda \mathbb{E}[g_a] - \delta_b\|_2^2 + \mathrm{tr}(\mathrm{Var}(g_b)) + \lambda^2 \mathrm{tr}(\mathrm{Var}(g_a)). \tag{B.2}$$

Differentiating w.r.t. $\lambda$ gives

$$\nabla_\lambda \mathbb{E}\big[\|\tilde{g} - g^*\|_2^2\big] = -2\langle \delta_b, \mu_a \rangle + 2\lambda \left(\|\mathbb{E}[g_a]\|_2^2 + \mathrm{tr}(\mathrm{Var}(g_a))\right), \tag{B.3}$$

which yields the minimizer

$$\lambda^* = \frac{\langle \delta_b, \mathbb{E}[g_a] \rangle}{\|\mathbb{E}[g_a]\|_2^2 + \mathrm{tr}(\mathrm{Var}(g_a))}. \tag{B.4}$$

In particular, under $\mathrm{Var}(g_a) = \sigma_a^2 I_d$, we have $\mathrm{tr}(\mathrm{Var}(g_a)) = d\sigma_a^2$.

**Practical approximation via dynamic analysis.** Eq. (B.4) can be rewritten as

$$\lambda^* = \frac{\|\delta_b\|_2}{\|\mu_a\|_2} \cdot \frac{\cos\theta}{1 + \rho}, \qquad \cos\theta := \frac{\langle \delta_b, \mu_a \rangle}{\|\delta_b\|_2 \|\mu_a\|_2}, \qquad \rho := \frac{d\sigma_a^2}{\|\mu_a\|_2^2}. \tag{B.5}$$

To obtain a computable rule, we parameterize the unobserved bias magnitude via the *relative bias ratio* $\beta := \|\delta_b\|_2 / \|\mu_b\|_2$ (with $\mu_b = \mathbb{E}[g_b]$) and treat $\beta \approx \kappa$ during the main training phase. Plugging $\|\delta_b\|_2 \approx \kappa \|\mu_b\|_2$ into (B.5) yields

$$\lambda^* \approx \eta \frac{\|\mu_b\|_2}{\|\mu_a\|_2}, \qquad \eta := \frac{\kappa \, c_\theta}{1 + \rho}. \tag{B.6}$$

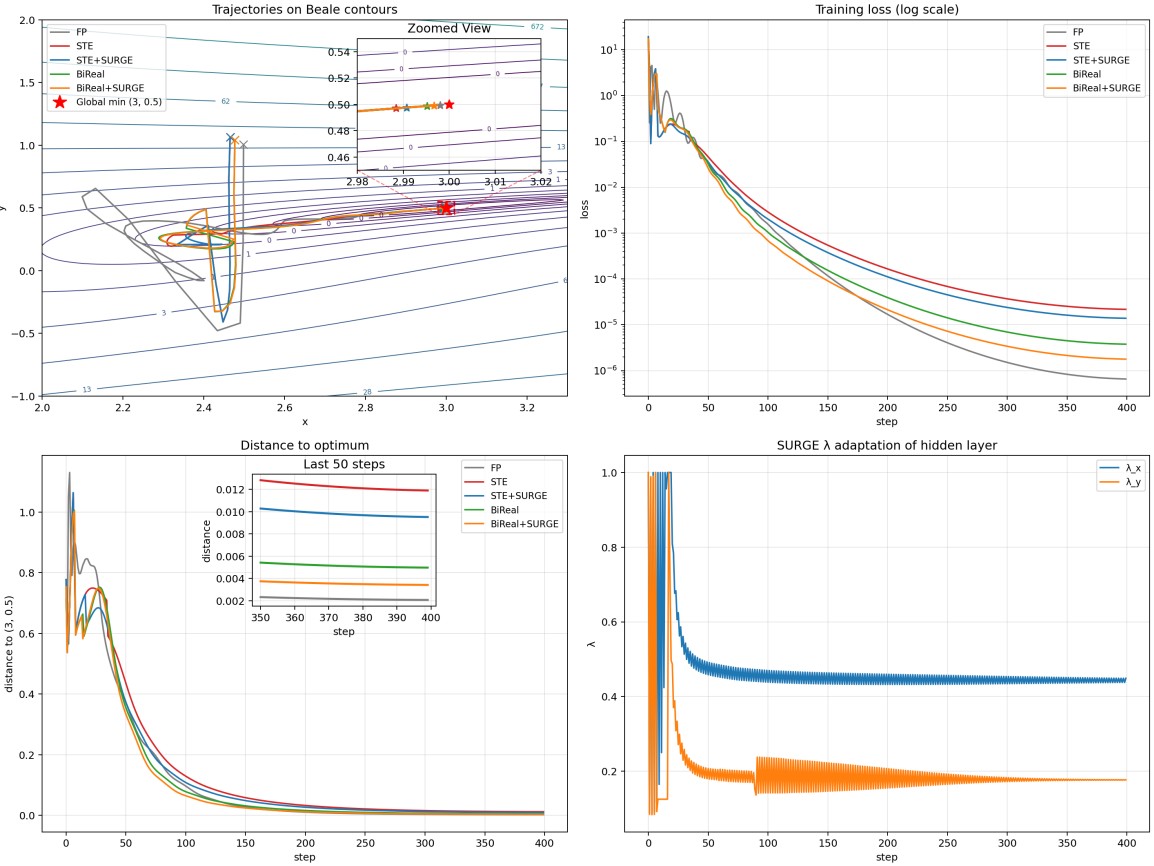

*Figure A.* Comparison of five methods (FP, STE, STE+SURGE, Bi-Real, Bi-Real+SURGE) on Beale function: (a) trajectories, (b) loss, (c) distance to optimum, (d) SURGE's $\lambda$ adaptation of hidden layer.

Finally, replacing population quantities by mini-batch estimates gives

$$\lambda_{\mathrm{AGS}} \approx \eta \, \frac{\|g_b\|_2}{\|g_a\|_2 + \epsilon}, \tag{B.7}$$

where $\epsilon > 0$ is a numerical stabilizer. Empirically, the inter-branch cosine similarity stays high over most of training and drops only near convergence (Figure F), while $\lambda$ quickly reaches a plateau in the toy study (Figure A), supporting the use of a slowly-varying $\eta$ in practice.

□

## C. SURGE for Toy Problem and Visualizations

We implement a simple yet illustrative toy model to optimize the non-convex Beale function. The original model architecture consists of an input layer, a hidden layer with ReLU activation, and an output layer that produces 2D coordinates. In our experiments, we binarize the first and second linear layer.

**Convergence and loss curve of toy model.** As illustrated in Figure A, we compare five training methods under identical initialization: FP (Full-precision network), STE, STE+SURGE, Bi-Real, Bi-Real+SURGE. We also provide a focused three-group subset (FP, STE, STE+SURGE) shown in Figure B for clearer visualization. Specifically, for each binarized layer (based on STE / Bi-Real), SURGE adds a parallel full-precision layer and merges their outputs. We provide trajectory plot, loss curve, distance to optimum, adaptive scaling factor ($\lambda$). It can be seen that SURGE achieves better convergence performance, yielding lower loss compared to the control group without SURGE integration. And the scale factor $\lambda$ also reaches convergence.

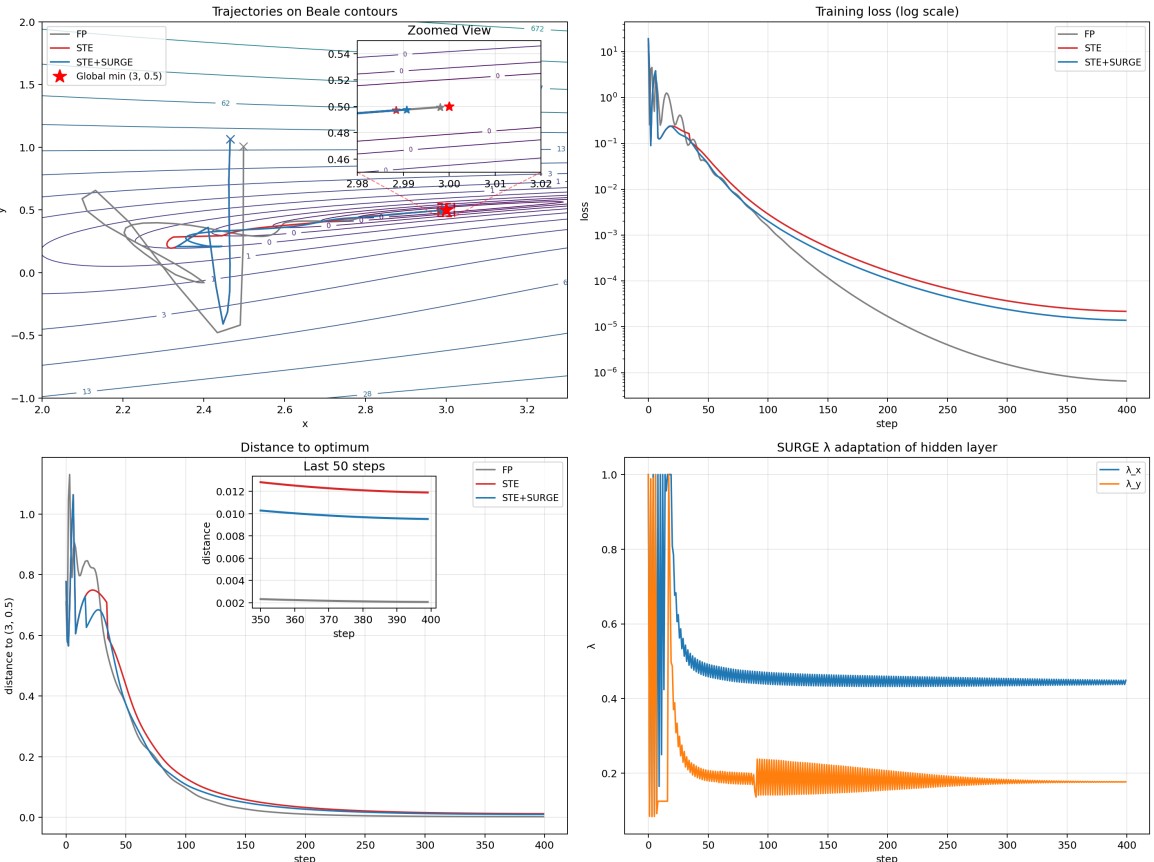

*Figure B.* Comparison of three methods (FP, STE, STE+SURGE) on Beale function: (a) trajectories, (b) loss, (c) distance to optimum, (d) SURGE's $\lambda$ adaptation of hidden layer.

**Parameter evolution of toy model.** As illustrated in Figure C, we provide a parameter evolution plot, where we track the Frobenius norms of the binary and auxiliary full-precision weights, as well as the learnable scaling factors $\alpha_w, \alpha_a$. We also provide a focused three-group subset (FP, STE, STE+SURGE) shown in Figure D for clearer visualization. As shown in the figure of Frobenius norm of weights, compared to the FP model, all binarized variants lie in a narrow band, but variants with SURGE maintain a slightly larger and more stable weight norm after the initial transient. As shown in the figure of learnable scaling factors, SURGE consistently pushes these scales slightly higher.

**Weight distribution.** As illustrated in Figure E, we provide the weight distribution of one layer in ResNet-18 trained with CIFAR-10. It can be seen that weights around zero is less with SURGE than the counterpart without SURGE. The binarization results without SURGE is less robust to any robust disturbance (Xu et al., 2022a), as sign(w) would more frequently flips.

**Cosine similarity.** As shown in Figure F, we draw a figure of cosine similarity (on ResNet-20 trained with CIFAR-10) between the gradient of weights of main branch and auxiliary branch, averaged over layers and mini-batches. It can be seen that the cosine similarity is relatively high. During the main training phase, the similarity slowly decreases but remains in the range 0.8-0.9. Towards the very end of training, when the model has already entered a small local basin and gradients become very small, the cosine similarity drops more sharply. This is expected: the full-precision branch can still perform fine-grained adjustments inside the basin, whereas the binary branch is constrained by quantization, so the auxiliary branch likely compensates in different directions.

**Noise contrast experiment.** As shown in Figure H, we conducted experiments with added noise on the toy model to more intuitively demonstrate that our compensation is not merely noise. The results show that the convergence process with added noise becomes significantly more volatile, and the final convergence performance is worse.

*Remark* C.1 (Phase-wise alignment and interpretation). The "approximately stable" alignment in Corollary 5.4 is intended to

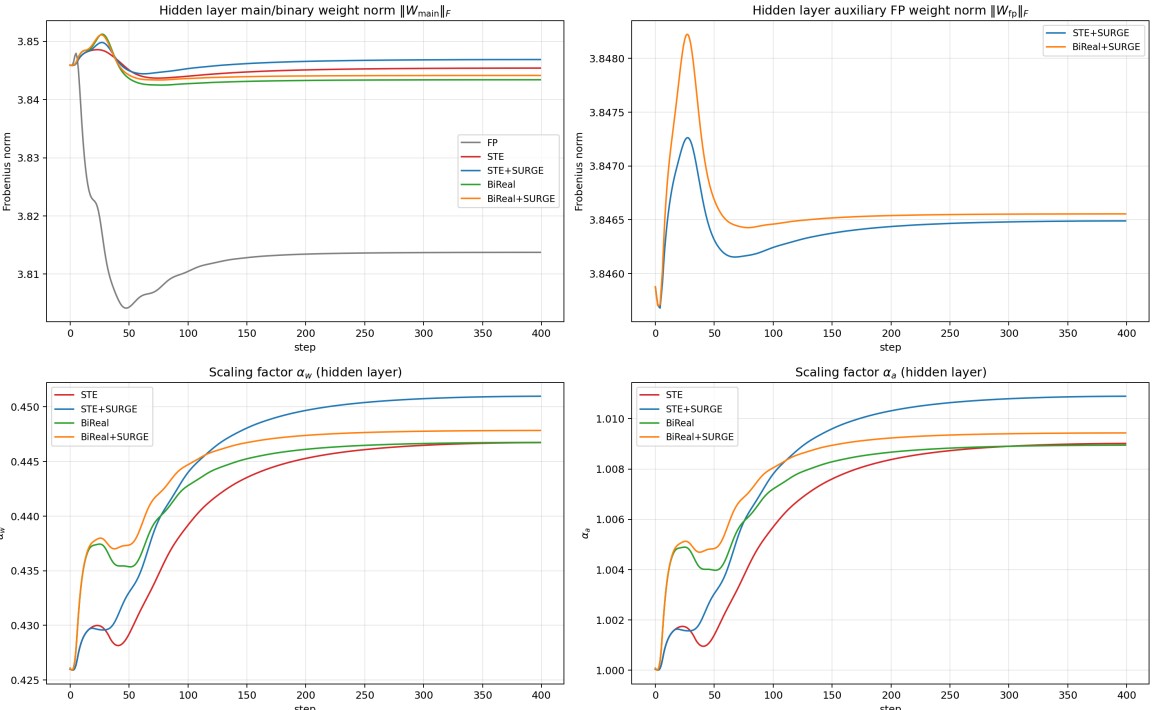

*Figure C.* Comparison of five methods (FP, STE, STE+SURGE, Bi-Real, Bi-Real+SURGE) on Beale function: (a) weight norm of main branch, (b) weight norm of auxiliary branch, (c) scaling factor of weights, (d) scaling factor of activations.

hold over the main optimization stage after warm-up. As observed in Figure F, the cosine similarity between the two branches remains high for most iterations, indicating that the auxiliary branch largely reinforces compatible descent directions. Near convergence, the similarity decreases as the binarized branch becomes more constrained within a quantization-induced basin, and the auxiliary (full-precision) gradient may play a more prominent role in directional correction and fine-grained refinement. We further observe a consistently positive cosine similarity between the gradients of inputs of the two branches in Figure G, suggesting that such compatibility also appears in the backpropagated layer-wise gradient signals. Empirically, the adaptive scale $\lambda$ quickly stabilizes after a brief transient (Figure A), supporting our "main-phase stability" approximation in Corollary 5.4.

## D. Dataset, Data Augmentation, and Evaluating Metrics

We evaluate on two standard image classification benchmarks, one object detection benchmark, and one suite of language understanding tasks to demonstrate the effectiveness: CIFAR-10 (Krizhevsky et al., 2009) (10k 32×32 images with random cropping & flipping), ImageNet-1K (Russakovsky et al., 2015) (1.28M training and 50k validation images at 224×224 resolution via center crop), PASCAL VOC (Everingham et al., 2010) (around 16k training and 5k validation images across 20 classes with multi-scale resizing to 1500×900, 1000×600, and 666×400, random flipping at 0.5 ratio), and GLUE (Wang et al., 2018) (covers CoLA, SST-2, MRPC, STS-B, QQP, MNLI (m/mm), QNLI, and RTE, without data augmentation).

We evaluate models using Top-1 and Top-5 accuracy for image classification, mean Average Precision at IoU=0.5 (mAP@0.5) for object detection, and task-specific GLUE metrics following the official protocol: Matthews correlation (MCC) for CoLA; accuracy for SST-2, MNLI (matched/mismatched), QNLI, and RTE; F1/Accuracy for MRPC and QQP; and Spearman correlations for STS-B.

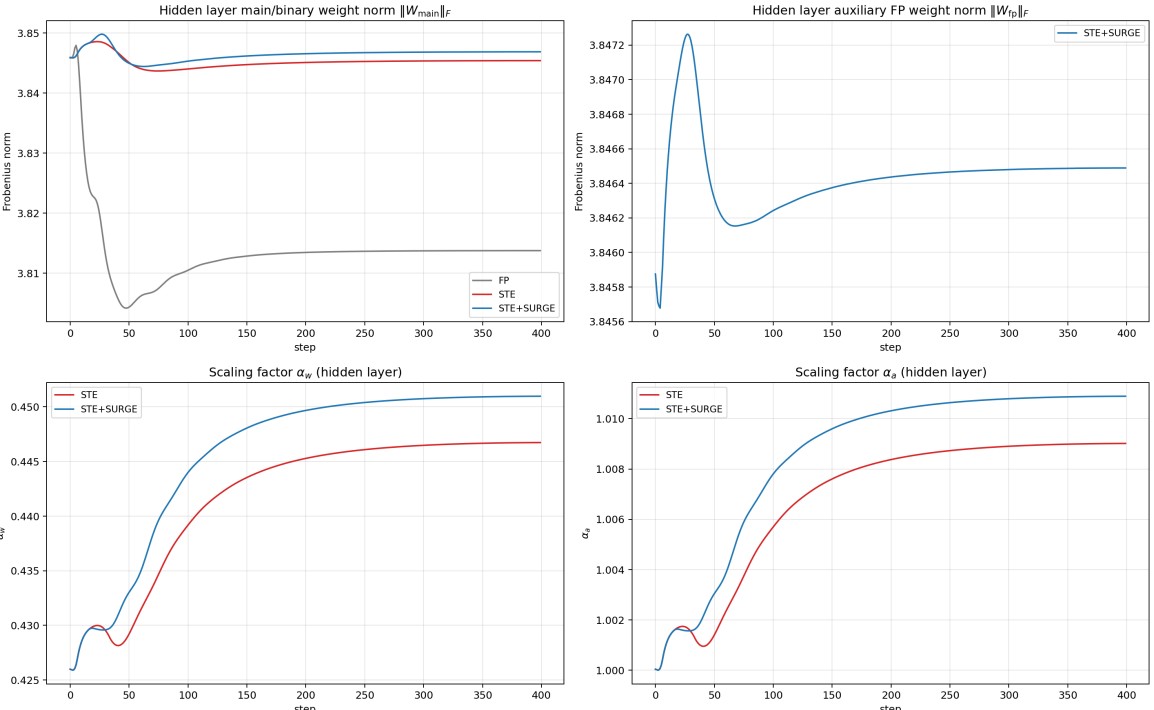

*Figure D.* Comparison of three methods (FP, STE, STE+SURGE) on Beale function: (a) weight norm of main branch, (b) weight norm of auxiliary branch, (c) scaling factor of weights, (d) scaling factor of activations.

## E. Implementation Details

### E.1. Model Details

**CIFAR-10.** On CIFAR-10, we evaluate our method with ResNet-18/20 (He et al., 2016) and VGG-Small (Simonyan & Zisserman, 2014). We binarize all convolutional and fully-connected layers except the first and last ones.

**ImageNet-1K.** On ImageNet-1K, we binarize ResNet-18 and retain the first layer, shortcut, and last layer in the networks as real-valued following (Liu et al., 2018b). We adopt the same model modification scheme as described in (Liu et al., 2020).

**PASCAL VOC.** On PASCAL VOC, we binarize Faster-RCNN with a ResNet-18 backbone. We keep the shortcut, first layer, and the last layer (the $1 \times 1$ convolution layer of RPN and an FC layer of the bbox head) in the detectors as real-valued after implementing 1-bit CNNs. Following (Wang et al., 2020), we modify the network of ResNet-18 with an extra shortcut and PReLU (He et al., 2015).

**GLUE.** On GLUE, we evaluate our method with BERT-base (Devlin et al., 2019). We follow the previous work to binarize the word embedding layer, MHA and FFN in transformer layers, but leave full-precision classifier, position embedding layer, and token type embedding layer (Qin et al., 2022; Liu et al., 2022).

### E.2. Training Details

**CIFAR-10.** On CIFAR-10, we train our models from scratch and following the setting in (Xu et al., 2021c), and the base scaling coefficient $\eta$ is set to $0.01$.

**ImageNet-1K.** On ImageNet-1K, we follow two implementation setups for fair comparison. First, we employ **one-stage training** on ResNet-18 following the setting in (Xu et al., 2022a), using Adam as the optimizer and a weight decay of $1e - 5$. The initial learning rate is set to $5e - 4$. The model is trained from scratch for 200 epochs with learning rates optimized by the annealing cosine learning rate schedule. Second, we employ **two-stage training** following the setting in (Liu et al., 2020), using Adam as the optimizer. The network is supervised by a real-valued ResNet-34 teacher. In the first stage, the model is trained from scratch with binarized activation and real-valued convolution weights. We load the state dict from the

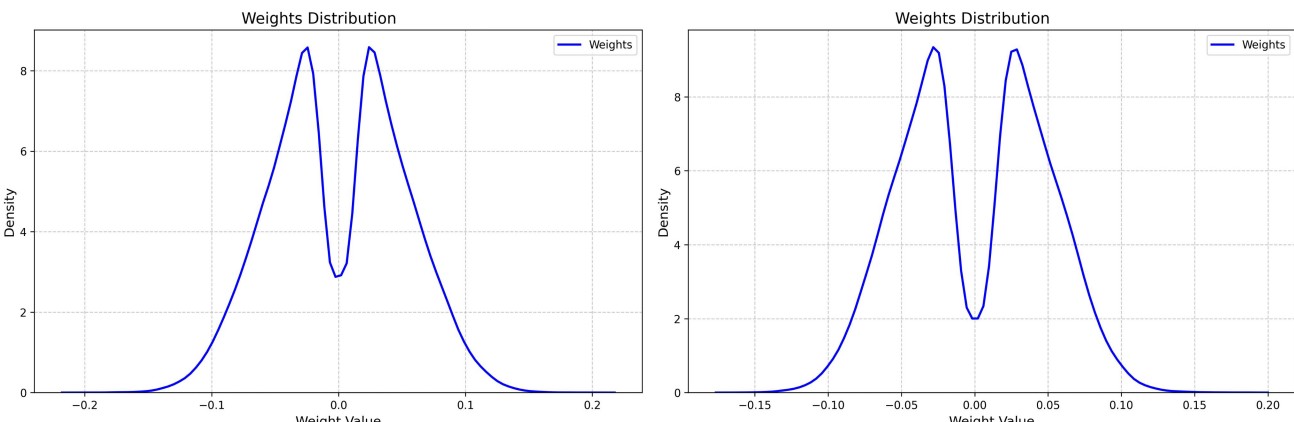

*Figure E.* Weight distribution comparison of a ResNet-18 layer trained on CIFAR-10. (left) Baseline method; (right) SURGE (ours).

first stage, and both the activation and weights are binarized in the second stage. The initial learning rate is set to $5e - 4$, the same as one-stage training, and annealed to 0 by a linear descent scheduler. The base scaling coefficient $\eta$ is set to $0.001$.

**PASCAL VOC.** On PASCAL VOC, we use ImageNet to pre-train the backbone of a 1-bit student, following (Liu et al., 2020). The SGD optimizer is utilized, and the batch size is set as 4 for Faster-RCNN. We train the model in two stages. Only the backbone is binarized at the first stage. Then we binarize all layers in the second stage. Each stage counts 12 epochs. The learning rate is set as 0.004 and decays by multiplying 0.1 in the 9th and 11th epochs following (Xu et al., 2022b). The base scaling coefficient $\eta$ is set to $0.001$.

**GLUE.** On GLUE, We follow (Liu et al., 2022) in adopting the experimental setting of (Devlin et al., 2019). We use the Adam as our optimizer, and we take more training epochs for every quantization method on each tasks to have a sufficient training, which is 50 for CoLA, 20 for MRPC, STS-B and RTE, 10 for SST-2 and QNLI, 5 for MNLI and QQP. We distill binary models using full-precision teacher without using multi-distill technique.

Furthermore, *we have provided our code* in the supplementary materials, which contains the full implementation of our method and training scripts to facilitate easy replication and future research.

# F. Additional GLUE Analyses

### F.1. Training under Fixed Wall-Clock Budgets

To further examine whether the improvement of SURGE comes from more effective optimization rather than simply longer training, we additionally evaluate BiT and BiT+SURGE under the same fixed wall-clock budget. Specifically, for each GLUE task, both methods are trained on identical hardware with the same task-specific training duration, and we compare the performance reached within that budget.

As shown in Table A, BiT+SURGE still outperforms the BiT baseline under the same time budget, improving the average GLUE score from 65.2 to 66.7. This indicates that, despite its higher per-step cost, SURGE provides more effective optimization updates within a controlled training-time budget.

*Table A.* Results under fixed wall-clock budgets on GLUE. For each task, both methods are trained on the same hardware with the same task-specific training duration.

| Method | MNLI-m (10800s) | MNLI-mm (10800s) | QQP (7200s) | QNLI (10800s) | SST-2 (7200s) | CoLA (3600s) | STS-B (1800s) | MRPC (1500s) | RTE (900s) | Avg. |
|---|---|---|---|---|---|---|---|---|---|---|
| BiT | 72.0 | 72.7 | 82.7 | 81.5 | 85.6 | 23.0 | 45.0 | 74.8 | 57.0 | 65.2 |
| BiT+SURGE | 72.1 | 72.3 | 83.3 | 82.4 | 86.5 | 21.4 | 54.3 | 78.7 | 54.5 | **66.7** |

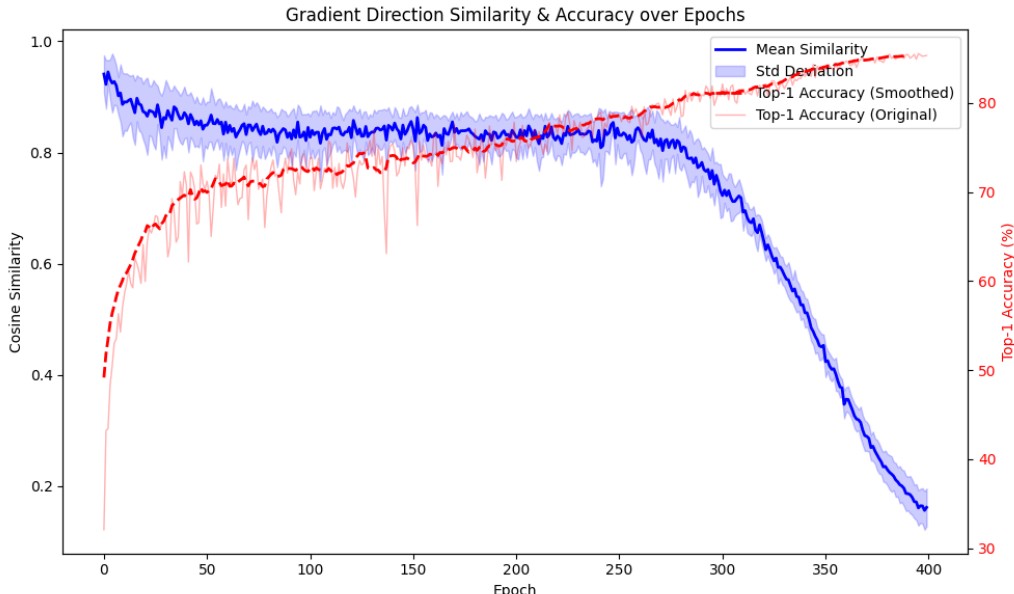

*Figure F.* Cosine similarity (on ResNet-20 trained with CIFAR-10) between the gradient of weights of main branch and auxiliary branch, averaged over layers and mini-batches

### F.2. Effect of Explicit Auxiliary-Branch Alignment

SURGE uses an auxiliary full-precision branch to compensate for the truncated first-order gradient caused by binarization, while the detach trick prevents the auxiliary branch from changing the forward output value. Therefore, the auxiliary branch is designed to provide complementary gradient information, rather than to mimic the binary branch in the forward computation.

We further test a variant that explicitly encourages the auxiliary branch output to align with the binary branch output. As shown in Table B, this explicit alignment does not further improve the performance over BiT+SURGE. This suggests that forcing the auxiliary branch to behave too similarly to the binary branch may reduce the diversity of the compensation signal, thereby weakening its ability to correct the STE-induced gradient bias.

*Table B.* GLUE dev-set results for BiT, BiT+SURGE, and BiT+SURGE+Align. All values are reported in percentage form.

| Method | MNLI-m | MNLI-mm | QQP | QNLI | SST-2 | CoLA | STS-B | MRPC | RTE | Avg. |
|---|---|---|---|---|---|---|---|---|---|---|
| BiT | 77.0 | 77.5 | 85.4 | 85.5 | 87.8 | 23.6 | 68.0 | 79.4 | 58.1 | 70.6 |
| BiT+SURGE | 77.3 | 77.5 | 87.1 | 86.2 | 88.6 | 24.1 | 71.7 | 80.6 | 60.6 | **72.0** |
| BiT+SURGE+Align | 77.1 | 77.3 | 85.9 | 85.9 | 87.8 | 23.6 | 71.3 | 79.2 | 58.8 | 71.2 |

## G. Overhead and Deployment Efficiency

SURGE introduces modest additional overhead during training while eliminating any extra inference cost, since the auxiliary branch is discarded after training.

### G.1. Training overhead.

**CNNs.** We conduct a comparison of training time (10 epochs) and memory (batchsize 256/GPU) among SURGE, Bi-Real Net, ReActNet, and RBONN under one-stage training on ImageNet. As shown in Table C, our results demonstrate that while the full-precision branch introduces modest overhead (+25% training time, +7.6% memory vs RBONN), SURGE can deliver significant accuracy improvements against other SOTA methods (+0.63% accuracy vs RBONN).

**Transformers.** We conduct a comparison of training time (1 epoch) and memory consumption between SURGE and

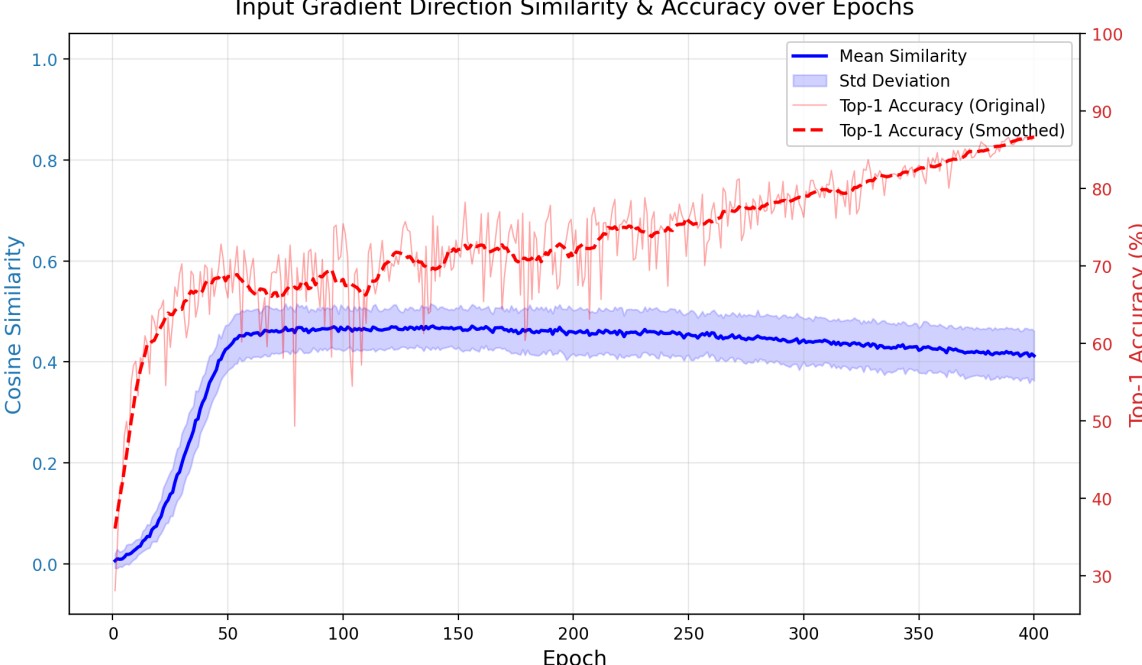

*Figure G.* Cosine similarity (on ResNet-20 trained with CIFAR-10) between the gradient of inputs of main branch and auxiliary branch, averaged over layers and mini-batches

*Table C.* Overhead comparison of training time (10 epochs) and memory (batchsize 256/GPU). Accuracy is reported after 10 epochs of training. * denotes a simple cost-reducing variant.

| Method | Training Time (min) | GPU Memory (2 GPUs) | Accuracy (%) |
|---|---|---|---|
| Bi-Real Net | 143 | 19153 MiB×2 | 38.73 |
| ReActNet | 156 | 20923 MiB×2 | 45.86 |
| RBONN | 160 | 21005 MiB×2 | 46.65 |
| **SURGE** | 200 | 22597 MiB×2 | **47.28** |
| **SURGE\*** | 177 | 22295 MiB×2 | 47.21 |

baseline (BiT) for BERT quantization on each task of GLUE. Following BiT, we employ task-specific batch sizes during training to optimize performance across different tasks. As shown in Table D, SURGE introduces acceptable additional training overhead (+17% avg time, +22% avg memory) while delivering significant accuracy improvements (+1.38% avg accuracy). Notably, when the baseline employs larger batch sizes (32) with substantial memory consumption (12429 MB) on QQP dataset, SURGE adds only minimal additional overhead (+10% memory).

Current BNN deployments predominantly target edge devices where inference efficiency is significant. Consequently, state-of-the-art methods in this domain prioritize two key metrics: (1) achievable accuracy under extreme quantization constraints, and (2) real-world inference latency on resource-limited hardware. Training efficiency remains secondary in established BNN research paradigms. Our method introduces modest additional overhead during training and does not impact deployment.

### G.2. Deployment efficiency.

SURGE discards all auxiliary branches after training, maintaining identical resource requirements to standard binary networks while delivering stable accuracy gains. Based on our prior experience, we implement the 1-bit models on ODROID C4, which has a 2.016 GHz 64-bit quad-core ARM Cortex-A55. By evaluating its real speed in real-world mobile device, the deployment efficiency of SURGE is proven. We leverage the SIMD instruction SSHL on ARM NEON to make the inference library BOLT (Feng, 2021) compatible with SURGE. We compare SURGE to the real-valued backbone in Table E.

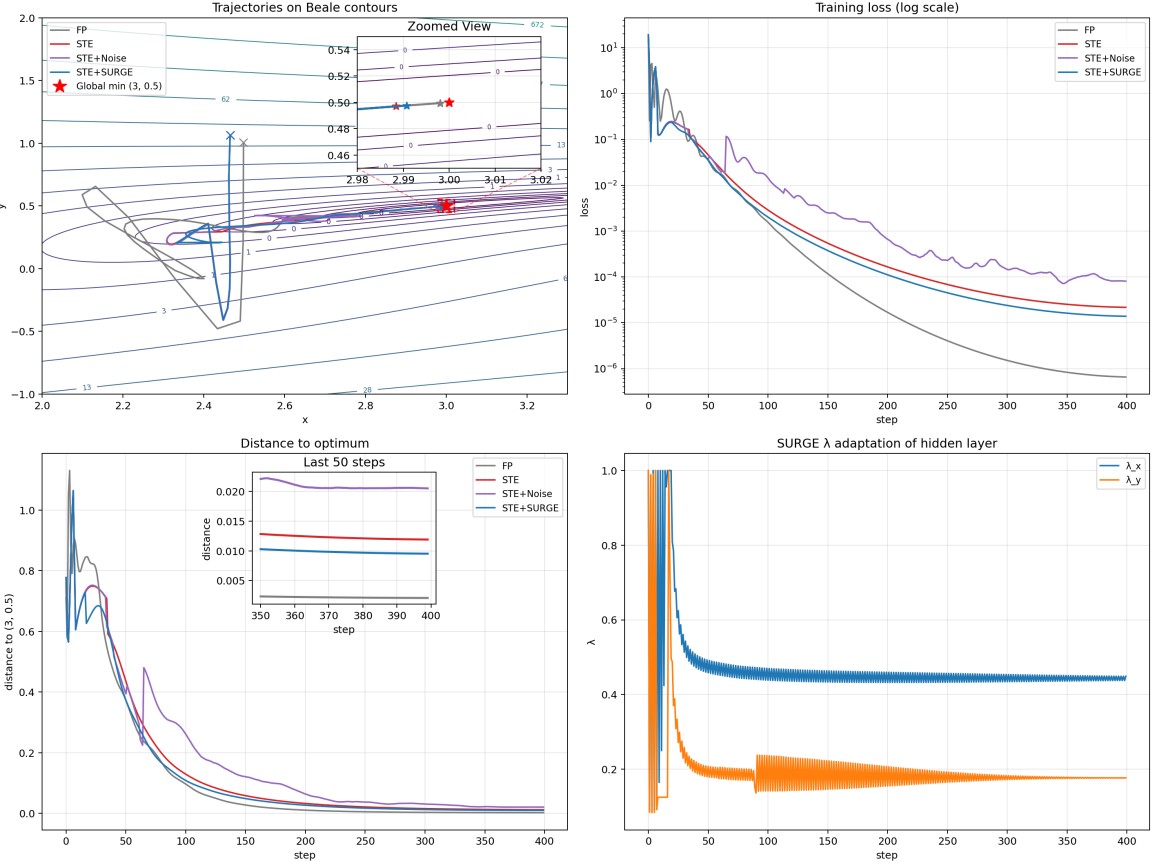

*Figure H.* Comparison of five methods (FP, STE, STE+Noise, STE+SURGE) on Beale function: (a) trajectories, (b) loss, (c) distance to optimum, (d) SURGE's $\lambda$ adaptation of hidden layer.

We can see that SURGE's inference speed is substantially faster with the highly efficient BOLT library. For example, the acceleration rate achieves about 4.1× on ResNet18.

## H. Limitations and Future Works

While SURGE improves gradient estimation through its dual-path design, this approach inherently requires the retention of auxiliary full-precision parameters $W_a^{(l)}$ throughout the training phase to enable gradient compensation. This architectural choice introduces two practical considerations: (1) a temporary increase of parameter memory footprint during backward propagation compared to conventional BNN implementations, and (2) additional computational overhead from parallel path gradient calculations during optimization. In Appendix G, we have conduct experiments to quantify the training overhead of binarizing CNNs and Transformers. Notably, these costs are strictly confined to the training phase. During inference, the auxiliary branches are discarded, restoring the original binary architecture's computational efficiency and memory footprint without any residual overhead.

For architecture, our framework enables exploration of auxiliary structure designs (*e.g.*, via efficient structure design, low-rank decomposition, or saliency-aware layer compensation) to reduce computational overhead. Here we propose a potential method to decrease training overhead. By simply replacing auxiliary convolutions from 3×3 to 1×1 kernels (SURGE* variant in Table C), we achieve 14.4% training time reduction (177min vs 200min over 160min), 1.5% memory savings (22295MB vs 22597MB over 21005MB), while still bringing accuracy improvements against other SOTA methods (+0.56% accuracy vs RBONN). This validates that our framework inherently supports efficient re-engineering, and such architectural explorations constitute promising future directions.

*Table D.* Comparison of Time, Memory, and Final Accuracy between Baseline and SURGE of training binarized BERT across GLUE tasks

| | **Method** | **CoLA** | **MNLI (m/mm)** | **MRPC** | **QNLI** | **QQP** | **RTE** | **SST-2** | **STS-B** |
|---|---|---|---|---|---|---|---|---|---|
| Batch size | ——— | 16 | 16 | 8 | 8 | 32 | 8 | 8 | 8 |
| Time (1 epoch)/s | Baseline | 107 | 5025 | 92 | 2590 | 3263 | 61 | 1583 | 144 |
| | SURGE | 122 | 6220 | 109 | 3050 | 4110 | 72 | 1755 | 158 |
| Memory (MB) | Baseline | 5701 | 8175 | 6051 | 6051 | 12429 | 6051 | 4617 | 6049 |
| | SURGE | 7227 | 9649 | 7475 | 7483 | 13731 | 7475 | 5957 | 7473 |
| Final accuracy | Baseline | 23.56 | 77.05/77.46 | 79.41 | 85.48 | 85.40 | 58.12 | 87.84 | 67.97 |
| | SURGE | 24.11 | 77.27/77.53 | 80.64 | 86.23 | 87.12 | 60.65 | 88.65 | 71.70 |

*Table E.* Deployment efficiency.

| **Backbone** | **Method** | **#bit (W/A)** | **Size (MB)** | **Memory Saving** | **Latency (ms)** | **Acceleration Rate** |
|---|---|---|---|---|---|---|
| ResNet-18 | Real-valued | 32/32 | 42.7 | – | 276.8 | – |
| | **SURGE** | **1/1** | **1.7** | **25.1×** | **67.8** | **4.1×** |

