# OpenReview forum: "SURGE: Surrogate Gradient Adaptation in Binary Neural Networks"
_ICML.cc/2026/Conference — ICML 2026 regular_

### Official Review · Reviewer_U5V5 · 2026-03-06

**Soundness:** 2
**Presentation:** 3
**Significance:** 2
**Originality:** 1
**Overall Recommendation:** 2
**Confidence:** 4

**Summary:**

This paper propose SURGE for improving STE. Specifically, DPGC adds a full-precison counterpart with identical dimensions as binary operator, using the stop gradient method to ensure only the gradient is modified and the forward path remains the same. AGS balances the original gradient from STE and the gradient from the additional counterpart by norm-based scaling. The proposed method equipped on CNNs are evaluated image classification tasks and object detection tasks. The proposed method equipped on Transformers are evaluated through language understanding tasks.

**Compliance With Llm Reviewing Policy:**

Affirmed.

**Final Justification:**

After the rebuttal, the main concern on the novelty of the paper is not addressed. The paper only proposed incremental modifications in my view. Although the paper is pretty clear, I recommend reject.

**Key Questions For Authors:**

1. What is the effect of the method? In figure 1 (b), what's the merit of more gradients? How does figure 1 (c) and (d) indicate the effect of the proposed methods?
2. What is the explicit form of $g_a$? How does this related with mitigated mismatch and reduced bias?

**Limitations:**

yes

**Strengths And Weaknesses:**

Strengths:
1. The motivation part of the paper is well-formulated and easy to follow.
2. The method is evaluated across CNN and transformer architecture on image classification, object detection and language understanding tasks.

Weaknesses:
1. This paper does not propose principled method. The detach trick which establishes an auxiliary path of the gradient is common when implementing custom gradient. The main contribution of this paper would be introducing an extra full-precison counterpart with same shape as binary operator, but the benefit is not clear.
2. The main claim and key improvement over STE is not clearly discussed. It is claimed that gradient mismatch is mitigated and bias of gradient is reduced but the related discussion is lacked.
3. The scalability of the introduced method is unknown. Complex manipulation over gradient increase the burden of QAT. As the model size increases, such methods become not practical. It seems like the proposed method doubles the trainable parameters. Time and memory consumption overhead compared with STE are required.

---

> ### Author Rebuttal · Authors · 2026-03-30
>
> We thank the reviewer for the feedback. We address the concerns point by point below.
>
> **(1) Weakness 1**
> **R1:** The detach operation itself is not our paper's novelty. Our contribution is instead the training-time dual-path compensation framework built on top of it: for each binarized operator, we introduce a full-precision auxiliary counterpart together with the Adaptive Gradient Scaler (AGS), so that the forward output remains identical to the original BNN, while the backward pass receives an additional compensating signal. The auxiliary branch is removed after training, so the method introduces no inference-time overhead. This “same forward / improved backward” design, together with norm-based adaptive balancing between the two gradient paths, is the core contribution rather than the detach trick alone.
>
> **(2) Weakness 2 / Question 1**
> **R2:** The key issue with STE is that, for activations, it uses a clipped first-order surrogate, resulting gradient and bias. Eq. (4) makes this explicit. Our auxiliary path provides a compensatory signal for the clipped first-order gradients from the Straight-Through Estimator (STE). Fig. 1 is intended to visualize exactly this point: SURGE restores part of the gradients and shifts the gradient distribution toward more informative nonzero magnitudes. Thus, the merit of Fig. 1(b) is not merely “having more gradients,” but recovering useful gradient information that STE removes; likewise, Fig. 1(c)(d) show that the distribution is less concentrated around clipped/vanishing values. This is also consistent with prior observations that retaining a less quantized gradient route can provide less biased gradient estimates under quantization [1].
>
> This interpretation is further supported by the ablation and the toy study. In Table 6(b), compensating only clipped gradients already improves over the baseline, while compensating all gradients performs best, which is consistent with mitigating clipping-induced mismatch. In Appendix C, the toy Beale-function experiments show that SURGE achieves better convergence and lower loss than the corresponding STE baseline, while the noise-control experiment shows that simply adding noise does not reproduce this effect and instead leads to more volatile optimization and worse final convergence. This indicates that the auxiliary path is providing a structured correction, not random perturbation.
>
> **(3) Weakness 3**
> **R3:** We agree that the scalability is an important concern. As shown in Appendix F, we have provided a comprehensive comparison of the time and memory consumption for both CNNs and Transformers. In terms of training overhead, SURGE introduces acceptable additional overhead during training. Specifically, for CNNs, Appendix F reports modest extra training time / memory relative to strong BNN baselines; for Transformers, the average overhead is about +17% training time and +22% memory, with an average accuracy gain of +1.4% on GLUE. Critically, this overhead ratio is not fixed. As noted in Appendix F, the relative overhead decreases with larger batch sizes because the fixed activation memory dominates. For example, on QQP with batch size 32, SURGE's memory overhead drops to only +10%, while yielding a significant improvement in accuracy (from 85.40 to 87.12). In terms of deployment efficiency, SURGE discards all auxiliary branches after training, maintaining identical resource requirements to standard binary networks while delivering stable accuracy gains. It is also worth noting that, in Sec. G, we discuss the exploration of efficient auxiliary structure designs to reduce computational overhead, which presents considerable optimization potential. We have explored a simple yet effective cost reduction variant (see Table A, Appendix F), which reduces training time by ~14% and memory by ~1.5% while retaining most of the performance gain (47.21% vs. 47.28% top-1 on ImageNet).
>
> **(4) Question 2**
> **R4:** Eq. (6) gives the combined backward signal as $\frac{\partial L}{\partial x} = g_b + \lambda g_a$, where $g_b$ is the STE-based gradient from the binary path and $g_a$ is the backpropagated gradient from the full-precision auxiliary path. In the simplified linear notation of the paper, $f_a(x) = W_a^\top x$, so $g_a = \frac{\partial L}{\partial f_a}\frac{\partial f_a}{\partial x}$, and analogously for convolution / attention projections in practice. Thus, $g_a$ is the standard backward signal induced by the full-precision auxiliary operator. Its role is to provide a less biased compensating signal than the STE branch, which helps mitigate mismatch introduced by STE’s clipped first-order approximation, while AGS keeps this correction controlled so that the binary-path gradient remains dominant. Empirically, the cosine similarity between the main and auxiliary branch gradients stays relatively high during most of training, indicating that auxiliary path usually provides a compatible correction direction rather than random perturbation.

---

> > ### Author Rebuttal · Reviewer_U5V5 · 2026-04-02
> >
> > “Same forward / changed backward” design is exactly what detach trick brings, I don't see anything conceptually new.
> >
> > In the proposed method, the forward and backward path still differ and I don't see any manual design to make them close, why is the mismatch mitigated? Methods like DSQ mitigate forward and backward mismatch, how is the DPGC backward gradient matches zero gradient of sign function?
> >
> > Gradient clipping in STE stablize training, more gradients don't translate to better training performance. In other words, why is the clipped gradient useful?
> >
> > STE gives identical mapping of gradient, what exactly do the authors mean by bias?
> >
> > [1] Differentiable Soft Quantization: Bridging Full-Precision and Low-Bit Neural Networks, ICCV 2019

---

> > > ### Author Response · Authors · 2026-04-03
> > >
> > > We appreciate your feedback and address each concern below.
> > >
> > > # Q1: Regarding "Same forward/changed backward"
> > > We respectfully disagree. We have never claimed `detach` as our novelty.
> > >
> > > Since the sign function is non-differentiable, existing methods typically train BNNs by modifying the quantizer in a similar manner: keeping binarization in the forward pass, while designing a **heuristic** gradient approximation in the backward pass (i.e., same forward/changed backward). For instance, Bi-Real Net [1] uses a hand-crafted piecewise polynomial, and DSQ [2] uses a parameterized sigmoid.
> > >
> > > **Instead, SURGE's core novelty is a fundamentally different paradigm**: rather than modifying the quantizer's internal gradient estimator with yet another hand-crafted function, we introduce a learnable full-precision auxiliary branch (DPGC) for principled, adaptive gradient compensation without manual design. Since our training paradigm introduces a new branch, we still need to ensure identical forward computation. Thus, the `detach` operation is **merely** an implementation detail to achieve this within our training paradigm. **In fact, the core of our method remains the auxiliary branch (DPGC), while its design also guarantees the "same forward/changed backward" property.**
> > >
> > > In summary, via output decomposition ($\text{output} = f_b - f_{ao}.\text{detach}() + f_{ao}$), the forward output stays identical to standard BNNs, while the auxiliary branch injects compensatory gradient signals. Furthermore, as another core contribution, AGS derives a theoretically optimal $\lambda^*$ (Theorem 5.3) to balance inter-branch gradients.
> > >
> > > # Q2 & Q4: Regarding "Gradient mismatch/bias"
> > > First, **"bias" refers to STE's systematic gradient estimation error** — a well-established **consensus**, not our invention. STE's identity mapping is inherently an inaccurate first-order approximation [3]. DSQ [2]: *"STE ignores the influence of quantization...its error will be amplified"*; FDA-BNN [4]: *"the approximation function they (STE, DSQ, etc.) used may corrupt the main direction of factual gradient"*; IR-Net [5]: *"discrete binarization always leads to inaccurate gradients and the wrong optimization direction"*; Quant-Noise [6]: *"STE introduces a bias in the gradients."* In essence, the inaccurate gradient estimation introduces a fundamental bias, a divergence between STE's approximated gradient and the "factual gradient" for optimizing BNNs [4]. Our formally defined ideal gradient $g^\*$ (Def. 5.1, Assumption B.1) captures precisely this concept — $\delta_b = g^* - \mathbb{E}[g_b]$ measures the gap, stemming from (1) identity-function approximation error and (2) fixed-range clipping discarding gradient information.
> > >
> > > Regarding DSQ: **SURGE operates in a different paradigm.** DSQ designs a smooth forward function to approximate sign, making the quantizer's forward/backward consistent. SURGE does **not** modify the quantizer. Instead, it compensates STE's error via an auxiliary full-precision gradient $g_a$, so $\tilde{g} = g_b + \lambda g_a$ better approximates the ideal $g^\*$ (Theorem 5.3). Empirically: (1) Fig. 1: SURGE recovers clipped gradients while keeping forward output unchanged, shifting the gradient distribution rightward; (2) Fig. F: cosine similarity between gradients of the two branches stays 0.8–0.9, confirming compatible corrections; (3) Fig. G: random noise causes worse convergence, confirming structured compensation.
> > >
> > > # Q3: Regarding "Gradient clipping"
> > > STE‘s `fixed-range` clipping is fundamentally heuristic, and prior studies have shown that applying fixed-range gradient clipping is suboptimal for training BNN. IR-Net [5] states: "the Clip approximation increases the difficulty for optimization and decreases the accuracy in practice. It is crucial to ensure enough updating possibility, especially during the beginning of the training process." **Furthermore, the adverse effect of clipping is already supported by our ablation and toy studies.**
> > > SURGE does not merely inject "more gradients" but recovers *useful optimization signals*:
> > >
> > > (1) **Table 6b**: compensating only clipped gradients ($|x|>1$): 87.4%→87.7%; only unclipped ($|x|\leq 1$): 87.6%; all: 88.0%. (2) **Fig. G**: random noise of similar magnitude causes volatile convergence and worse performance — SURGE's compensation is structured, not noise. (3) **Fixed wall-clock budget** (R5 for Reviewer 55q9): under same training time, SURGE outperforms baseline (GLUE avg 66.7 vs. 65.2), confirming improvement from gradient quality.
> > >
> > > # References
> > > [1] Bi-Real Net: Enhancing the Performance of 1-bit CNNs, ECCV 2018\
> > > [2] Differentiable Soft Quantization, ICCV 2019\
> > > [3] Bridging Discrete and Backpropagation, NeurIPS 2023\
> > > [4] Learning Frequency Domain Approximation for Binary Neural Networks, NeurIPS 2021\
> > > [5] Forward and Backward Information Retention for Accurate Binary Neural Networks, CVPR 2020\
> > > [6] Training with Quantization Noise for Extreme Model Compression, ICLR 2021.

---

### Official Review · Reviewer_55q9 · 2026-03-12

**Soundness:** 4
**Presentation:** 3
**Significance:** 3
**Originality:** 3
**Overall Recommendation:** 5
**Confidence:** 3

**Summary:**

The paper proposes SURGE, a novel method for training binary models. It relies on auxiliary layers that use full precision. Instead of relying solely on the Straight-Through Estimator (STE) to backpropagate gradients through the binary layer, the auxiliary layers are also used to refine the gradient approximation. The auxiliary layer’s gradient is weighted using Adaptive Gradient Scaler (AGS), which is theoretically motivated to minimize gradient error. Extensive experiments are performed on image classification, object detection and NLP.

**Compliance With Llm Reviewing Policy:**

Affirmed.

**Final Justification:**

The rebuttal addressed my concerns.

Additionally, the authors also experimented with the idea I suggested (question 2).

**Key Questions For Authors:**

1. What would happen if all the methods were trained using the same budget (say, a fixed duration)? Since SURGE has better quality gradient but is slower, would it still beat the other baselines?
2. Have the authors tried to align the auxiliary layer with the binary layer? Intuitively, the gradient would be approximated best if both layers did similar computations. Nothing enforces this in SURGE, but if the auxiliary layer was encouraged to match the binary layer’s output, it may lead to higher quality gradients.

**Limitations:**

yes

(Consider including the "limitations and future works" section -- currently in Appendix G -- to the main paper, or at least part of it. )

**Strengths And Weaknesses:**

### Strengths

- The paper is well written and easy to read. The core ideas and motivations are conveyed in a pedagogical manner.
- The problem tackled is very timely: binary models are extremely efficient models at inference, and the main difficulty is training them. SURGE attempts to close the gap between binary model training, and full precision training.
- While the choice for the scaling $\lambda_{AGS}$ is natural, the authors also back it up with theoretical insights (Theorem 5.3 and Corollary 5.4), which under some assumptions and approximations is the optimal scaling for minimizing gradient errors.
- I find the experimental section quite solid. The method is evaluated on a wide range of tasks (image classification, object detection, NLP) and models, and is compared each time against multiple baselines. The results are convincing, with SURGE consistently achieving the best score, although with marginal gains sometimes.
- Additional ablations are performed, further showcasing the necessity of each component.

Overall I find the method quite natural. It makes me think of other works where an auxiliary network with full expressivity is used to train a constrained model — for instance to train a sparse attention layer [1] (I am not an author of this paper). This idea appears in different contexts, and I find appealing to apply it to binary model training as well.

[1] *SSA: Sparse Sparse Attention by Aligning Full and Sparse Attention Outputs in Feature Space*, Shen et al., 2025

### Weaknesses

- In addition to the binary model, the method requires training in parallel a full-precision model, which does not benefit from the same speedup as its binary counterpart. This makes training quite inefficient. However I understand that: i) the training time and memory will remain similar to training a standard full-precision model, which is manageable, and ii) we may be mostly interested in inference speed rather than training speed, if the model is planned for some specific uses for instance. The authors also provide training time and memory in Appendix F, showing that SURGE is slower but remains in the similar order of magnitude as the other baselines.
- Gains are relatively small compared to the overhead induced by the method.

Minor remarks:

- In Section 3, notations for $W$ are  bit confusing. The authors talk about a “neural layer”, but $W$ is a vector instead of matrix. I assume the authors consider a single neuron?
- In Figure 2 (a), are residual connections missing from the transformer block?

---

> ### Author Rebuttal · Authors · 2026-03-30
>
> **We sincerely appreciate Reviewer 55q9’s valuable feedback and constructive comments. Our point-by-point responses are detailed below.**
>
> **(1) W1: Training Efficiency**\
> **R1:** We thank the reviewer for the constructive feedback and the comprehensive understanding of our proposed method. It is also worth noting that, in Sec. G, our framework enables exploration of efficient auxiliary structure designs (e.g., via efficient structure design, low-rank decomposition, or saliency-aware layer compensation) to reduce computational overhead, which presents considerable optimization potential. Such architectural explorations constitute promising future directions.
>
> **(2) W2: Gains**\
> **R2:** Thank you for your concern. It’s worthy to note that while the improvements may appear relatively small in some experiments, our method **consistently** enhances performance metrics across all tested scenarios. For instance, regarding BERT binarization on the GLUE benchmark. SURGE consistently improved the baseline BiT [1] by +1.4% average accuracy across all tasks. In terms of training overhead, our framework enables exploration of efficient auxiliary structure designs to reduce computational overhead, which presents considerable optimization potential. In terms of deployment efficiency, SURGE discards all auxiliary branches after training, maintaining identical resource requirements to standard binary networks while delivering stable accuracy gains.
>
> **(3) Minor Remark 1**\
> **R3:** Thank you for pointing this out. Denoting $W$ as a weight vector here is solely to simplify the mathematical derivation. In general, SURGE is fully applicable to linear layer, convolutional layer, and attention projection operators. This generalization is further elaborated in Fig. 2 and Sec. 4.1. We will revise the manuscript to make this distinction clearer.
>
> **(4) Minor Remark 2**\
> **R4:** Thank you for pointing this out. The residual connection was omitted solely to simplify the computational illustration. We will correct this in the revised manuscript.
>
> **(5) Q1**\
> **R5:** We thank the reviewer for this important question. Following the suggestion, we additionally evaluated all methods under the same **fixed wall-clock budget** rather than the same number of epochs. Concretely, for each GLUE task, we assigned the same task-specific training duration to all methods on identical hardware, and then compared the final performance reached within that budget.
>
> The results show that **SURGE still outperforms the STE baseline under the same time budget**, improving the average GLUE score from **65.2** to **66.7** (**+1.5**). This indicates that SURGE’s advantage does not come merely from training longer, but from providing more effective optimization updates even when total training time is controlled.
>
> | Method | Avg. |
> |--------|------|
> | bit / STE baseline | 65.2 |
> | bit\_SURGE         | 66.7 |
> | $\Delta$           | +1.5 |
>
> The full per-task fixed-budget results are provided in **https://anonymous.4open.science/r/25232-SURGE-3996/tabs/bit_fixed_duration.md**.
>
>
> **(6) Question 2**\
> **R6:** We appreciate the reviewer's suggestion. Our design of the auxiliary branch is specifically motivated by the need to directly compensate for the truncated first-order gradient of the STE, rather than to mimic the binary branch in the forward computation. For this reason, we do not explicitly align the auxiliary layer with the binary layer.
>
> To examine the reviewer’s hypothesis, we additionally tested a variant that encourages the auxiliary layer to align with the binary layer output. The table below (full table in https://anonymous.4open.science/r/25232-SURGE-3996/tabs/bit_align.md) suggests that such explicit alignment does not further improve performance. This is consistent with our intuition: forcing the auxiliary branch to behave too similarly to the binary branch may reduce the diversity of the compensation signal, thereby weakening its ability to correct the STE-induced gradient bias.
> | Method            | Avg. |
> |-------------------|-----:|
> | BiT               | 70.6 |
> | BiT+SURGE         | 72.0 |
> | BiT+SURGE+Align   | 71.2 |
>
> [1] Liu et al. 2022 BiT: Robustly Binarized Multi-distilled Transformer.

---

> > ### Author Rebuttal · Reviewer_55q9 · 2026-04-03
> >
> > I thank the authors for their thorough response. My concerns were addressed and I am increasing my score to 5.

---

### Official Review · Reviewer_wJv9 · 2026-03-12

**Soundness:** 3
**Presentation:** 3
**Significance:** 3
**Originality:** 3
**Overall Recommendation:** 5
**Confidence:** 3

**Summary:**

This paper proposes SURrogate GradiEnt Adaptation (SURGE), a novel training-time gradient compensation framework for Binary Neural Networks (BNNs). To mitigate the gradient mismatch issue caused by the Straight-Through Estimator (STE), SURGE introduces a Dual-Path Gradient Compensator (DPGC) that incorporates an auxiliary full-precision branch during training. This branch provides less biased gradient estimates without altering the forward pass or adding inference cost. To balance the contributions from the binary and auxiliary paths, an Adaptive Gradient Scaler (AGS) dynamically adjusts the scaling factor based on the norm ratio of the two gradient streams, which is derived from a theoretical analysis. Experiments across image classification (CIFAR-10, ImageNet), object detection (PASCAL VOC), and language understanding (GLUE) benchmarks demonstrate that SURGE achieves state-of-the-art results, notably achieving 62.0% top-1 accuracy on ImageNet with a one-stage trained binarized ResNet-18.

**Compliance With Llm Reviewing Policy:**

Affirmed.

**Key Questions For Authors:**

In the overhead analysis for Transformers (Table B, Appendix F), the baseline (BiT) uses different batch sizes per task. For QQP, the batch size is 32, leading to a high memory baseline of 12429 MB. Can you elaborate on why SURGE's overhead is "minimal" here (+10% memory), and is there a trade-off where SURGE requires a smaller batch size to stay within memory constraints for tasks with a smaller baseline? This would be relevant for practitioners with fixed GPU memory budgets.
The ablation on parameter
ηη (Figure 3) shows that fixed scaling is highly sensitive, while AGS is robust. However, the peak accuracy for AGS occurs at
η=0.01
η=0.01. How was this optimal
η η selected, and is it a sensitive hyperparameter across different datasets and architectures? Is the "short transient" period mentioned in Corollary 5.4 long enough to allow for adaptive tuning of this parameter?

The method is demonstrated primarily for weight binarization. Could SURGE be directly applied to binarize activations as well? Would the gradient compensation mechanism work similarly, or are there specific challenges when the auxiliary path is applied to the activation function?

**Limitations:**

Increased Training Cost: As acknowledged in Appendix F, SURGE introduces a non-trivial increase in training time (e.g., +25% for ResNet-18 on ImageNet) and memory usage compared to some prior BNN methods. While this is a trade-off for higher accuracy, it is a practical limitation for large-scale or resource-constrained training scenarios.

Extra Hyperparameter: The method introduces a new hyperparameter,
η
η, the base scaling coefficient. While the authors show the method is more robust to this than fixed scaling, the optimal value still needs to be determined, adding a small overhead to the model selection process.

Architectural Constraint: The method's design is predicated on the existence of a "binary path" and an "auxiliary path." This structure is inherently tied to a specific architectural pattern. Its application to more diverse or unconventional network architectures not based on this dual-path formulation is not discussed.

Theoretical Simplifications: The theoretical analysis, while insightful, relies on several strong simplifying assumptions (uncorrelated noise, isotropic noise, fixed bias ratios). The practical success of the method may not be fully explained by this simplified model, and the exact conditions under which the derived AGS rule is optimal remain to be fully verified in the complex, non-convex landscape of BNN training.

**Strengths And Weaknesses:**

Strengths:

Novel Approach to a Known Problem: The paper directly addresses the core challenge of gradient mismatch in BNN training. The dual-path compensation mechanism is a clever, training-only solution that introduces no inference overhead, a key requirement for efficient BNNs.

Strong Theoretical Motivation: The authors provide a theoretical foundation for their method, starting with a formal definition of gradient statistics, making assumptions about gradient noise, and deriving an optimal scaling factor for the auxiliary path (Theorem 5.3). This theoretical grounding leads to the practical AGS update rule (Corollary 5.4), lending credibility to the method's design.

Comprehensive Empirical Validation: The experimental evaluation is extensive and robust. The method is tested across diverse domains (vision and language), multiple network architectures (CNNs, Transformers), and various benchmarks. The reported results consistently show state-of-the-art performance, with significant improvements over prior methods like IR-Net and ReCU.

Clear Ablation Studies: The paper includes meaningful ablation studies that validate the individual contributions of the DPGC and AGS components. The study on the gradient compensation scope effectively shows that SURGE's design overcomes the limitations of fixed-range clipping in STE. The analysis of training overhead (Appendix F) provides important context for the method's practical utility, acknowledging a trade-off between training cost and final accuracy.

Weaknesses:

Incomplete Overhead Analysis: While Appendix F provides valuable training overhead comparisons, the analysis for Transformers is less detailed than for CNNs. For example, the increased memory consumption for tasks like QQP is noted but not discussed in the context of the method's design or whether this overhead can be optimized further. A more systematic analysis of the computational cost relative to the accuracy gain would strengthen the practical argument for the method.

Limited Analysis of the Auxiliary Path's Role: The paper posits that the auxiliary path provides "higher-order terms" to compensate for STE's bias. While the experimental results support this claim, there is no direct analysis (e.g., via visualization or gradient statistics) of what kind of information the auxiliary path provides. Comparing the gradients from the main and auxiliary paths could offer valuable insight into the mechanism of compensation.

Scope of Gradient Compensation: The method is presented as a general gradient compensator for the STE. However, the experiments are focused on weight binarization. The potential applicability and performance of SURGE for binarizing activations, or for jointly binarizing both weights and activations, is not explicitly explored or discussed as a separate experiment, even though it's a standard scenario in BNN literature.

---

> ### Author Rebuttal · Authors · 2026-03-30
>
> **We sincerely appreciate Reviewer wJv9’s valuable feedback and constructive comments. Our point-by-point responses are detailed below.**
>
> **(1) W1: Incomplete Overhead Analysis**
>
> **R1:** We thank the reviewer for highlighting this point. As shown in Table B, regarding the training overhead for Transformers, the absolute memory increase caused by auxiliary branch is stable and acceptable across all GLUE tasks (ranging from 1,302 MB to 1,520 MB). Importantly, the performance gains outweigh the computational costs. On the QQP dataset, for example, since the baseline already exhibits a substantial memory footprint, the relative increase in memory usage is marginal (from 12,429 MB to 13,731 MB, i.e., +10%), yet it yields a significant improvement in accuracy (from 85.40 to 87.12). Furthermore, consistent with the future work discussed in Sec. G, this additional overhead for Transformers can be further mitigated through techniques such as applying low-rank decomposition to the weights of the auxiliary branch. We will incorporate a systematic cost-gain analysis for Transformers and detail these potential optimization directions in our revision.
>
> **(2) W2: Limited Analysis of the Auxiliary Path's Role**
>
> **R2:** We thank the reviewer for this suggestion. Regarding the gradient information of the auxiliary path, we will provide a more rigorous analysis and explanation. The current manuscript contains direct evidence demonstrating that the auxiliary path provides a compensatory signal for the truncated first-order gradients by the Straight-Through Estimator (STE). Specifically, Fig. 1 shows that SURGE successfully recovers activation gradients outside the STE clipping range, shifting the gradient distribution towards more informative direction. Furthermore, Fig. F illustrates a high cosine similarity between the gradients of the main and auxiliary branches. This indicates that the auxiliary path typically provides corrective signals that are compatible with the primary optimization direction, rather than mere random noise. We agree that these findings should be more explicitly framed as a mechanistic analysis in the main text, and we will refine the corresponding discussions in our revision.
>
> **(3) W3: Scope of Gradient Compensation**
>
> **R3:** Yes. In fact, our reported BNN settings already binarize both weights and activations. The same DPGC/AGS principle applies to the activation-side STE as well: the auxiliary path provides a complementary backward signal where the clipped STE is most information-losing.
>
> **(4) Q1**
> **R4:** We appreciate the reviewer's questions. It is worth noting that we adopt the same batch size settings for each GLUE task as in BiT [1]. Since the majority of GPU memory in BNN training is consumed by the binarized branch, the additional overhead from SURGE's auxiliary branch is relatively small, resulting in a stable absolute memory increase across all tasks (as shown in Table B). Consequently, for the QQP dataset, because its baseline already exhibits a high memory footprint, the relative memory increase is minimal. Furthermore, as detailed in Appendix F, the average memory increase introduced by SURGE for Transformer binarization is 22%. This training memory increment is typically manageable. Therefore, there is generally no need to reduce the batch size in most cases. Admittedly, under extreme memory constraints, a trade-off does exist. In fact, training efficiency remains a secondary concern in established BNN research paradigms. Since SURGE discards all auxiliary branches after training, it maintains inference resource requirements identical to those of standard binary networks, while delivering stable accuracy gains.
>
> In our method, the base scaling coefficient $\eta$ can be rapidly determined. In practice, the calibration of $\eta$ employs empirical gradient norm profiling **within 10 initial iterations**. Specifically, we establish $\eta$ such that $\eta \cdot r \approx \mathcal{O}(1)$, where $r = \frac{\|g_b\|_2}{\|g_a\|_2 + \epsilon}$. This ensures the scaling factor $\lambda$ maintains an appropriate order of magnitude ([0.1, 10]) for stable gradient compensation. Consequently, we can consistently identify a suitable hyperparameter configuration across different datasets and architectures. The term "short transient" mentioned in Corollary 5.4 refers to the brief transitional period after which the adaptive scaling factor $\lambda$ of AGS enters a relatively stable phase. As illustrated in Figure A, $\lambda$ stabilizes rapidly and no extra adaptive tuning for $\eta$ is required.
>
> **(5) Q2**
> **R5:** Please see response to Weakness 3.
>
> [1] Liu et al. 2022 BiT: Robustly Binarized Multi-distilled Transformer.

---

### Official Review · Reviewer_z92X · 2026-03-18

**Soundness:** 2
**Presentation:** 2
**Significance:** 3
**Originality:** 3
**Overall Recommendation:** 3
**Confidence:** 3

**Summary:**

The paper proposes an alternate method to train binary neural networks where the weights and activations are all binary valued (+1/-1). Standard methods replace the weights and activations in the computational graph (say node A) by pair of nodes (A --> B) with B sitting in the place of A, and the edge connecting A --> B in the forward pass is simply the sign operation. However, as the sign operation is not differentiable, the "backward" routine for this sign function module is modified to just be the identity rather than the 0 vector (which is the correct gradient for the sign function for most inputs). While this already works and gives non-trivial performance, it begs the question of -  can the parameters be updated better? Especially, as gradient is viewed as the linear approximation for "infinitesimal changes", but in a discrete world infintesimal changes are impossible.

The paper proposes an alternate parameter update method that attempts to account for it using compensatory paths, and argues for why it is a better update method and supports its conclusion with a set of empirical results on standard supervised learning tasks on standard deep learning architectures (e.g. CIFAR10/Imagenet1k on Resnet etc).

**Compliance With Llm Reviewing Policy:**

Affirmed.

**Final Justification:**

Theoretically, the ideal role aimed for by the auxiliary path is not made clear.

Empirically it looks like there maybe inconsistent comparison with baselines (e.g. the RBNN accuracy numbers for CIFAR10 are taken directly from the paper whn the full-precision workflow seems to have changed from that paper).

Both are significant concerns. So I vote to not accept the paper at the current stage.

The idea has potential, and the empirical results seem positive, and the problem being tackled is important. So I am okay with it being accepted if my two concerns above are addressed.

Hence, I am sticking to a 3 out of 6 for now.

**Key Questions For Authors:**

Points 2,4,5 and 6 on soundness in the weakness section are all questions for the authors.

**Limitations:**

Not applicable.

**Strengths And Weaknesses:**

Strengths:

The motivation for why the BNN is unsatisfactory is good, and it has wide ranging collection of empirical results supporting the validity of the method.

Weaknesses:

1. Clarity: Several issues seem to be unaddressed/left ambiguous in the description of the algorithm. E.g. the two weight vectors W_a and W_b of the auxilliary and main paths -- how are they related throughout training? How are they initialised? The way they are used suggests they are both real valued parameters, but W_b enters the model only through Q_W(W_b). Are they somehow tied to each other (say through a sign operation W_b = sign(W_a)) every time W_a changes? Is W_b even changing directly? The parameters W_a and W_b are completely absent in the main figure (Figure 2).

   * The algorithm in the appendix makes some of these questions clear, but it needs to be made explicit in the figure/text itself.

2. Soundness: Assuming the algorithm A in the appendix is right -- you maintain two different real valued weight vectors W_a and W_b. How are they supposed to be related? As W_a is completely irrelevant in the test phase, I assumed W_b is such that it somehow approximates W_a, but they are not even initialised to be equal -- I am failing to see how they are related. And if W_a and W_b are completely unrelated, why would the gradient along the auxiliary path W_a even be helpful for updating W_b? Isn't it possible that they act at cross purposes? Also, W_a does seem to affect the backprop, but not the forward prop so technically it can get really large in magnitude how is this handled?

   * I can see that these are all questions the authors have also had -- Figures C and D would be the first things to try to assuage these concerns. But I am still not fully convinced how having a separate W_a (not explicitly tied to W_b) can help in a better way to update W_b (which is actually the only thing that matters in the inference phase).

   * I appreciate the attempt at illustrating the algorithm as shown in Figures A, B and G for a synthetic problem. I really do believe that it is possible the proposed algorithm (or a variant) could be doing intelligent updates for BNNs, but convincing me would require a detailed breakup of the various components of the algorithm at least for the toy problem considered in Figures B and G.

    * Also, what exactly is being plotted in the trajectory plots (Figure A,B,G top-left)? Is it the 2d predicted vector for some chosen input as the parameter evolves through training?


3. Convention: The partial derivative notation has a fixed meaning. You can have alternate objects that mimic the gradient and these could be the update steps taken by the parameters, but redefining the partial derivative is not something any of us is free to do. One needs to get permission from Newton and Leibniz to do so. This throws me off in several places.


4. Soundness: The statements in Assumption 5.2 and Theorem 5.3 and Corrollary 5.4 seem artificial and unsound. For example what exactly is meant by "reference gradient"? The exact gradient can be computed exactly (it is mostly 0), but just useless for optimisation. How is this "reference" gradient supposed to fix this? This needs elaboration.

    * One possiblity I can think of is that, when restricting the parameters to take values in the d-dimensional hypercube {+1,-1}^d, the role of this modified "gradient" is to say what the loss would become if you move from the current vertex of the hypercube to any neighbouring vertex -- this corresponds to extending the notion "linear approximation over nearby points" that the gradient gives in the real numbers setting to the discrete setting corresponding to the hypercube.

     *  But the paper barely talks about what is the "desired property" of the "reference gradient" approximation that using an auxiliary path is supposed to get you. Assumption B1 in the appendix is a good start -- but how this surrogate affects anything else in the rest of the paper is not clear. Mainly how does using auxiliary paths get better access to this object?


5. Soundness: Using \delta_b from the assumption in Theorem 5.3 seems like cheating. In no reasonable world would you have access to the deviation of the measured gradient from a "reference gradient". You might as well assume you are given the reference gradient itself.

6. Soundness: I am anti squiggle equals in Theorem/corollary/lemma statements. They have their place in proof sketches or the text of the paper. Corollary 5.4 uses them liberally. It gives no sense of when the approximation holds and how good it is even when it holds.

---

> ### Author Rebuttal · Authors · 2026-03-30
>
> We thank the reviewer for the careful reading and for identifying several places where the current presentation can be made more explicit and more carefully scoped.
>
> **(1) W1 / W3**
>
> **R1:** In our implementation, both $W_b$ and $W_a$ are real-valued trainable tensors; $W_b$ is updated directly; $W_a$ and $W_b$ are not tied by a sign constraint; and only $Q_W(W_b)$ participates in the forward value of the binary branch. $W_b$ is initialized from the corresponding full-precision pretrained model, whereas $W_a$ is randomly initialized.We will revise the manuscript accordingly.
>
> **(2) W2: Relation between $W_a$ and $W_b$, and why the auxiliary path can help**
>
> **R2:** The key point is that the usefulness of $W_a$ does not rely on $W_a \approx W_b$ in parameter space. The two branches are attached to the same layer, receive the same layer input, and are **optimized under the same loss**, but they play different roles: $W_b$ parameterizes the actual binary branch used at inference, whereas $W_a$ exists only to provide an auxiliary backward signal. What matters is whether the auxiliary branch provides complementary layer-level descent information to the STE-based gradient on the same optimization objective. Appendix C shows that the cosine similarity between the main and auxiliary gradients remains generally high during the main training phase (Fig. F; about 0.8–0.9), suggesting that the auxiliary path usually provides a compatible correction rather than acting at cross purposes.
>
> Concretely, DPGC keeps the forward exactly unchanged and only augments the backward signal through $g_b + \lambda g_a$. More broadly, this is in line with prior observations that retaining a less quantized gradient route can be beneficial when purely STE-based updates become severely biased under quantization [1].
>
> Regarding the reviewer’s concern about the auxiliary magnitude, this is precisely one of the issues that our second component, AGS, is designed to address. By dynamically rescaling the auxiliary contribution based on the observed gradient norms, AGS prevents the auxiliary path from overwhelming the update and helps maintain stable training.
>
> We also added a component-wise breakdown on the toy problem in Figures B/G. The full SURGE remains the best-performing one. (https://anonymous.4open.science/r/25232-SURGE-3996/imgs/toy_ablation/FigureB_ablation.png and https://anonymous.4open.science/r/25232-SURGE-3996/imgs/toy_ablation/FigureG_ablation.png)
>
> [1] Stock et al. Training with Quantization Noise for Extreme Model Compression, ICLR2021
>
> **(3) W4: What is the “reference gradient”?**
>
> **R3:** Thank the reviewer for raising this point. In our paper, $g^\*$ is not the exact derivative of the hard sign function. Rather, it is an oracle surrogate target: the population gradient induced by the best smooth surrogate $s^\*$ within a surrogate family (Appendix B.1). The purpose of introducing $g^\*$ is to formalize a target notion of a “better surrogate signal” than a fixed STE-style backward rule, not to claim access to the true sign gradient.
>
> Under this interpretation, Theorem 5.3 is meant to study whether the combined signal $g_b + \lambda g_a$ can be viewed as moving closer to such an oracle surrogate target, because $g_a$ carries full-precision information not subject to STE clipping. We agree that this role should have been stated much more explicitly, and we will move the definition of $g^\*$ into the main text and label it clearly as an oracle analysis target.
>
> **(4) W5: Use of $\delta_b$**
>
> **R4:** We agree that $\delta_b$ is not observable in practice. Our intent was not to assume access to $\delta_b$ during training. It appears only in the oracle analysis to motivate the form of the scaling rule. The actual algorithm never uses $\delta_b$ or $g^*$; Algorithm A uses only observable quantities and updates $\lambda_{\mathrm{AGS}} = \eta \frac{\|g_b\|_2}{\|g_a\|_2+\epsilon}$. We will revise Sec. 5 to make the oracle-to-practical transition explicit, so that Theorem 5.3 is not read as an implementable assumption but as motivation for the practical AGS form.
>
> **(5) W6: Use of “$\approx$” in a corollary**
>
> **R5:** We agree with this criticism. As written, Corollary 5.4 overstates the degree of formality. Our actual intent was to present a practical approximation / motivation for the AGS norm-ratio rule, not a theorem-level guarantee with an explicit error bound. In the revision, we will remove the “$\approx$” statements and present them as a practical approximation remark, with its intended regime of use stated explicitly. The algorithm itself does not depend on that approximation being interpreted as a uniform formal guarantee.
>
> Overall, Sec. 5 is intended as principled guidance for the method design, rather than a complete formal guarantee in its current form. It motivates the construction of the auxiliary path and AGS, while the main contribution is the resulting practically effective mechanism. We will revise it accordingly.

---

> > ### Author Rebuttal · Reviewer_z92X · 2026-04-04
> >
> > Thanks for the rebuttal.
> >
> > W1, W3: The presentation concerns are still significant. Partial derivatives have a fixed definition from basic calculus. The issue with the sign-quantisation is not that partial derivatives become undefined or difficult to obtain. It is just that they become useless. So we need to use something else other than gradients. The paper is about comparing different alternatives, and NOT about redefining the partial derivative. The paper proposes SURGE which is better than SWE (at least empirically). The appendix pseudocode also has significant presentation issues -- it requires significant context and not easily implementable for a reader. e.g how does the g_a and g_b terms even affect the parameter updates g_{wb} and g_{wa}? How is $\lambda$ different from $\lambda^{l,t}$?
> >
> > W2: This is the most crucial concern. I understand that W_a and W_b are not directly linked and are not scalar multiples of each other. But what is the desired goal of W_a? Assume you had all the information about the loss landscape, what would the ideal W_a be for any given W_b? Can you show that the W_b and W_a update will ensure that this is maintained throughout training? (at least approximately). The experiments on cosine similarity between W_a and W_b indicate that you desire them to be proportional. That however can't be the only goal, what other objective is desired out of W_a?
> >
> > W4: I can see the definition of reference gradient in the appendix, but I don't see how it makes an impact at all in the main paper. For example, if we replaced the definition of "reference gradient" in the appendix by "a vector of random numbers", how would it affect any of the theoretical arguments in the main paper?
> >
> > These weaknesses are not easily fixed. However, the main advantage of this paper is the (small but) consistent empirical advantage over the comparison algorithms. So it maybe important to show the hyperparameter selection protocol, to just show that the improved performance is not just due to more hyperparameters being tried out.
> >
> > I will still keep my rating at 3, as my primary theoretical concerns are still unaddressed.
> >
> >
> >
> > **The below is to be read after the "reply rebuttal comment" below. I am adding it to my Rebuttal acknowledgement instead so that the authors can see it as well.**
> >
> >
> > Thanks for the reply.
> >
> > W2: I still do not see an answer to the question "Assume you had all the information about the loss landscape, what would the ideal W_a be for any given W_b?". You can use all information of the loss landscape including the gradients for answering this question. I would like to see at least a property that the "ideal W_a" should satisfy. What is a "useful" gradient? Seems unclear.
> >
> > W4: The fact that g^* is the reference gradient as defined in the appendix doesn't seem to be used in the Theorems/corollaries or discussions in the paper.
> >
> > Below is a new issue. I had not noticed this while reviewing. But it seems, the empirical comparisons may be incomplete/unfair.
> >
> > I had to read the RBNN paper (Neurips 2020) for some other reason and noticed that the Resnet-18 numbers for CIFAR-10 in that paper was 92.2%. This matches the numbers in the current SURGE submission under RBNN in Table 1. However, in the RBNN paper the full-precision is reported as having 93% accuracy, but in the current SURGE submission the full precision is reported as having 94.8% accuracy. There could be multiple reasons for this, maybe it is just random seed variation, or maybe they found a better data augmentation/preprocessing etc.
> >
> > But if such changes were done to the full-precision workflow, it should also be done to all the comparison baselines. It is not fair to take the 92.2% number directly from the Neurips 2020 RBNN paper-- which might have used worse preprocessing or something else.
> >
> > I took a look at the supplementary, and they do not provide details on how all the numbers in the table are arrived at. Only the details for arriving at the SURGE numbers is given.
> >
> > Maybe I am missing something. If possible (and allowed by ICML/Openreview) let me know how this can be squared.

---

> > > ### Author Response · Authors · 2026-04-07
> > >
> > > We sincerely thank the reviewer for the feedback. We address each point below.
> > >
> > > # W1/W3
> > > Our use of $\partial$ with $\approx$ (Eq. 3) denotes the STE's surrogate for backward gradient, which is consistent with prior BNN works (ReCU Eq. 5, IR-Net Eq. 15). We will revise our formulation for clarification.
> > >
> > > $\lambda$ in Sec. 5 is the generic scale. $\lambda^{(l,t)}$ in Algorithm A is per-layer, per-iteration: step $t$ uses $\lambda^{(l,t-1)}$ in forward, then updates $\lambda^{(l,t)}=\eta\|g_b\|_2/(\|g_a\|_2+\epsilon)$ after backward.
> > >
> > > $g_a, g_b$ are **input gradients** ($\partial L/\partial x^{(l)} = g_b + \lambda g_a$, propagated to previous layers); $g_{wa}, g_{wb}$ are **weight gradients** updating $W_a, W_b$ (Alg. A, lines 21–22).
> > >
> > > # W2
> > > **Clarification.** First, we would like to clarify a potential misunderstanding regarding the cosine similarity experiments. Figure F measures the cosine similarity between the weight **gradients** ($g_{wa}$ and $g_{wb}$), not the weights themselves ($W_a$ and $W_b$). The desired role of $W_a$ will be explained in detail below.
> > >
> > > **The desired role of $W_a$: gradient compensation, not weight alignment.** Our work starts with the concept of gradient compensation. Specifically, STE‘s two heuristic designs (identity approximation and clipping) motivate a correction for gradient bias/wrong direction (as stated in prior work FDA-BNN). Therefore, we propose an auxiliary branch with full-precision parameter $W_a$ that involves no sign or STE, whose **sole purpose** is to provide an adaptive gradient compensation. While $W_b$ is used in the binary forward pass, $W_a$ only shapes the backward signal and is discarded after training. The two have no explicit coupling. They share the same loss and layer input, but play fundamentally different roles.
> > >
> > > **Ideal $W_a$.** The benefit of $W_a$ manifests not through the forward pass but through the backward pass: $g_a$ propagates to previous layers as part of $\partial L/\partial x^{(l)} = g_b + \lambda g_a$. There is no closed-form mapping from $W_b$ to an optimal $W_a$, because one governs a forward computation and the other provides a backward correction. This is why we formulate the optimal criterion by gradient (Theorem 5.3: $\min_\lambda \mathbb{E}\|g_b + \lambda g_a - g^\*\|^2$), defining what "good compensation" means without prescribing what $W_a$ should look like in weight space. Since $W_a$'s specific form is not fixed, any auxiliary-branch architecture that produces a useful $g_a$ is valid. A cost-reduced variant (SURGE*, Appendix G) demonstrates this flexibility.
> > >
> > > **Evidence that gradient adaptation works:**
> > > - Noise ablation on CIFAR-10: we conduct an experiment replacing $g_a$ with matched-norm Gaussian noise which yields 86.5% ($\leq$ baseline 87.4%); learned $g_a$ achieves 88.0%, showing that $g_a$'s **direction** matters. The mechanism is consistent with gains on ImageNet/VOC/GLUE.
> > > - Gradient similarity of layer input (https://anonymous.4open.science/r/25232-SURGE-3996/imgs/input_grad_cosine.png): $g_a$ and $g_b$ are not fully aligned ($\cos(g_b, g_a) \approx 0.45$), allowing $g_a$ to offer directional corrections to $g_b$.
> > > - Table 6b: compensating only clipped gradients ($|x|>1$, where $g_b=0$) improves 87.4%→87.7%, showing $g_a$'s contribution is complementary.
> > > - On the toy model of Appendix G, where oracle gradient ($\partial L/\partial y \cdot W_\text{latent}$, no sign) is computable, $\cos(g_\mathrm{SURGE}, g_\mathrm{oracle}) \approx 0.95 > \cos(g_\mathrm{STE}, g_\mathrm{oracle}) \approx 0.87$, with lower loss (https://anonymous.4open.science/r/25232-SURGE-3996/imgs/oracle_gradient_alignment.png).
> > >
> > > # W4
> > > If $g^\*$ were replaced by a random vector, Theorem 5.3 would no longer characterize compensation toward a meaningful reference gradient; it would instead reduce to fitting an arbitrary target. In that sense, the theoretical interpretation would collapse.
> > >
> > > For example, under a zero-mean random reference $r$ uncorrelated with $g_a$ and $g_b$, the optimal coefficient becomes $\lambda^\*=\frac{\mathbb{E}\langle r-g_b,\;g_a\rangle}{\mathbb{E}\|g_a\|^2}.$ Since $\mathbb{E}[\langle r,g_a\rangle]=0$, this reduces to $\lambda^\*=-\frac{\mathbb{E}\langle g_b,\;g_a\rangle}{\mathbb{E}\|g_a\|^2},$ which is negative when $g_a$ and $g_b$ are positively aligned, as empirically observed. Thus, the optimal coefficient becomes negative in expectation rather than a positive compensation coefficient. This illustrates that the analysis relies on $g^\*$ being a structured reference, not arbitrary noise.
> > >
> > > # Hyperparameters
> > > SURGE only adds **one** hyperparameter $\eta$. Our selection protocol only depends on a simple measurement: in the first 10 iterations, compute $r=\|g_b\|_2/(\|g_a\|_2+\epsilon)$ and set $\eta$ so $\lambda = \eta \cdot r \in [0.1, 10]$. This yields $\eta=0.01$ for CIFAR-10 and $\eta=0.001$ for ImageNet/VOC/GLUE. Moreover, in Fig. 3b, all experiments with $\eta\in[0.005, 0.013]$ outperform baseline.

---

### Decision · Program_Chairs · 2026-04-30

**Decision:**

Accept (regular)

**Comment:**

This work introduces SURGE, a training-time gradient compensation framework for binary neural networks that uses an auxiliary full-precision branch to provide less biased gradient estimates than the Straight-Through Estimator (STE), combined with an adaptive gradient scaler (AGS) to balance the two gradient streams.

Reviewers were split on this submission, with two rejects (one weak) and two accepts.

Reviewers 55q9 and wJv9 recommended acceptance, citing the clear writing, broad experimental coverage (vision + language), and the theoretical motivation. Reviewer 55q9 noted the method feels natural and that the rebuttal addressed their concerns. Unfortunately, Reviewer wJv9 did not engage with the rebuttal (no acknowledgement) or participate in the post-rebuttal discussion, which is critical for borderline papers.

Reviewer z92X (with the most detailed technical engagement) raised several soundness issues with the theoretical framework: the role of the auxiliary path relative to the binary path is not made precise, the "reference gradient" in Assumption 5.2 and Theorem 5.3, and the use of approximate equalities in formal theorem statements (Corollary 5.4). Reading the detailed rebuttal of the authors, I found these concerns to ultimately be minor misunderstandings. Reviewer U5V5 found the contribution incremental, arguing that the detach trick for custom gradients is standard and the benefit of the full-precision counterpart is not clearly established. I disagree and think t hat the experiments do establish benefit of the method, although I agree the detach trick is standard in the literature. I side with the authors regarding the detach trick; it is not their intended contribution so it does not matter that the detach trick is a standard tool in this domain.

During the post-rebuttal discussion, I asked all reviewers to engage with Reviewer z92X's concerns. However, only Reviewer z92X responded. They voiced their concern about the fairness of empirical comparisons: the RBNN baseline numbers in Table 1 appear to be taken directly from the original 2020 paper, but the full-precision accuracy reported in this submission (94.8%) is noticeably higher than in the RBNN paper (93%), suggesting different preprocessing or training setups; as was repeated in their final justification. The authors clarified to me that
- all CIFAR-10/ResNet-18 numbers in their Table 1, including both the baseline methods and the FP reference, are taken directly from ReCU (ICLR 2022, Table 4)
- no selective modification was done and this will be made explicit in the paper.

Thus I do not find this to be grounds for rejecting the paper. In light of this, I am recommending to accept the paper.